# Is Layer Normalization Fine-tuning Sufficient for Visual Distribution Shifts?

## Abstract

Layer Normalization (LayerNorm) is crucial in the functionality of Vision Transformers Foundation models (ViTFs), yet its role in fine-tuning under data scarcity and domain shifts remains underexplored. Our study reveals that LayerNorm parameter shifts (LayerNorm shifts) after fine-tuning are key indicators of a model's adaptation from a source to a target domain. The adaptation's success relies on how well the target domain's true distribution is represented by the training samples. These insights provide a theoretical foundation for connecting LayerNorm shifts with domain shifts. Building on these insights, we introduce the Fine-tuning Shift Ratio (FSR) to quantify representation consistency and propose an innovative rescaling mechanism using a scalar ($\lambda$), inversely related to FSR. This aligns LayerNorm shifts with optimal data representation conditions and includes a cyclic framework to improve fine-tuning. Extensive experiments across various datasets and settings validate our approach. In Out-of-Pretraining (OOP) tasks, lower FSR and higher $\lambda$ highlight under-represented training samples, while ViTFs tuned for In-Pretraining (IP) scenarios favor conservative updates. Our findings illuminate LayerNorm dynamics in transfer learning, offering practical fine-tuning strategies.

## 1 Introduction

Vision Transformers (ViTs) (Dosovitskiy et al., 2020), along with their variants, lead visual foundation models (ViTFs) by incorporating Layer Normalization (LayerNorm) before or after attention layers (Radford et al., 2021; Oquab et al., 2023; Lu et al., 2024; Wang et al., 2024; Ding et al., 2024). While ViTFs excel in transferring across downstream tasks as pretrained backbones, full-model fine-tuning is resource-intensive and inefficient (Jie & Deng, 2023; De Min et al., 2023; Zhao et al., 2023). Fine-tuning solely the LayerNorm parameters with predictors (LayerNorm fine-tuning) (Sandler et al., 2023) offers a competitive alternative. Studies like (Zhao et al., 2023; Chen et al., 2024; De Min et al., 2023; Qi et al., 2022; Giannou et al., 2023) show its effectiveness, yielding competitive/superior results with minimal model updates. Notably, (Giannou et al., 2023) explores normalization layers' expressive power, showing that fine-tuning these layers alone in random ReLU networks can accurately replicate any target network $O\sqrt{width}$ times smaller. However, the mechanisms of LayerNorm fine-tuning in ViTFs remain underexplored, raising our primary research question: ***Can LayerNorm fine-tuning effectively handle data distribution shifts in visual foundation models across various domains and sample sizes? How can its adaptability be improved?***

For clarity, the standard definition of LayerNorm is provided as follows:

$$ln(*) = \left((z - \mathrm{E}[z])/\sqrt{\mathrm{Var}[z] + \epsilon}\right) \odot \gamma + \beta. \tag{1}$$

where $z$ is the input variable; $\gamma, \beta$ are learnable scale and bias, and $\epsilon$ is a small value to prevent numerical problems. After the LayerNorm operation, $ln(z) \sim \mathbb{P}_{\beta,\gamma^2}$, where $\beta$ is the variance mean and $\gamma^2$ is the variance; $\mathbb{P}_{\beta,\gamma^2}$ is an unspecified distribution.

This paper systematically investigates LayerNorm's role in ViFs' fine-tuning, focusing on adaptation in both In-Pretraining (IP) and Out-of-Pretraining (OOP) scenarios (Teterwak et al., 2024) across various data scales like one-shot, few-shot, and full fine-tuning. To gain deeper insights into the mechanisms of LayerNorm fine-tuning, the study explores LayerNorm fine-tuning by using controlled toy experiments to manage variables like distributional shifts and data conditions, then validates

findings with large-scale pretrained visual models on datasets of natural and pathological images. Our experiments uncovered the following key findings:

- **LayerNorm fine-tuning can bridge the gap in data shift if the target training data accurately represents the target domain, especially under source domain influence (Section 2).** Specifically, convergence of LayerNorm parameters depends on the training data size, with $\gamma$ needing more data to converge than $\beta$ (Section 2.1).

- **Tuning LayerNorm with predictors might hinder its ability to capture distributional shifts,** as it can entangle data and target shifts, disturbing LayerNorm's intended function. (Proposition 3.1 in Section 3)

- In practical applications involving both natural and pathological images, **LayerNorm fine-tuning can _overshoot_ during IP tasks, but usually _undershoots_ in OOP tasks if there are enough fine-tuning samples**[1].

To address the insights derived from our findings, we propose targeted solutions. Rescaling $\gamma$ shifts with a non-negative scalar $\lambda$ can further improve performance by aligning better with the target domain. Then, we propose a robust tuning framework that alternates between training the predictor and fine-tuning LayerNorm layers, enabling them to better capture data shifts and consistently improve performance across different training sample regimes. Furthermore, our empirical evaluations suggest a fine-tuning strategy where $\lambda < 1$ is optimal for IP fine-tuning, while $\lambda > 1$ for OOP scenarios to enhance adaptability. The codebase, data splits, and resources will be publicly released upon acceptance to enable reproducibility and further research.

## 2 RELATIONSHIP BETWEEN LAYERNORM SHIFTS AND DOMAIN SHIFTS

**Notations.** A standard ViT consists of $n$ blocks, each typically containing two LayerNorm layers: $\mathcal{M} = \{enc, [m_1', ln_1^1, m_1, ln_1^2], \ldots, [m_n', ln_n^1, m_n, ln_n^2], output\ layers\}$, where $m'$ and $m$ denote possible attention or other layers; the two LayerNorms is defined by $ln^1$ and $ln^2$. Superscripts $S, T$, and $T*$ refer to source, target training, and ideal target domain variables. Datasets are indicated by $(X^S, Y^S)$ for source data and labels, $(X^T, Y^T)$ for target training data and labels, and $(X^{T*}, Y^{T*})$ for all target data. Model states are $\mathcal{M}^S$ for the original pretrained model, $\mathcal{M}^T$ for the tuned model, and $\mathcal{M}^{T*}$ for the ideally tuned model. Predictive layers $\mathcal{C}$ are usually added at the end of $\mathcal{M}$ for classification. To prevent effects from $\mathcal{C}$, the paper uses lossless linear layers for $\mathcal{C}$ in classification tasks. For simplification, we denote all LayerNorm layers $\mathcal{LN}$ in $\mathcal{M}^S$ $\mathcal{M}^T$ and $\mathcal{M}^{T*}$ as $\mathcal{LN}^S$ $\mathcal{LN}^T$ and $\mathcal{LN}^{T*}$.

**Assumptions.** 1. For self-supervised ViTF inputs $X^S$, there exists a corresponding supervised $Y^S$ that aligns with the target task labeled data $(X^T, Y^T)$; 2. ViTFs can extract useful features from $X$ without fine-tuning, although such extraction may be suboptimal; 3. A predictor $\mathcal{C}$ exists that can map these features to $Y$. These are generally held for large-scale pre-trained vision transformers.

**LayerNorm shifts encode data shifts.** This part shows that only LayerNorm layers in the model capture data distributions while the rest layers remain static.

**Proposition 2.1.** _Under the assumption that the $Shift_{ln}, Shift_{data}$ functions are proportionally comparable across domains. This implies that the magnitude of change in LayerNorm parameters tracks the magnitude of data distribution shift. Although not universally true, it is a reasonable approximation for theoretical analysis and is supported empirically. The shift from original to fine-tuned LayerNorm layers is proportional to the data distribution shift between $X^S, X^T$._

$$|Shift_{ln}(\mathcal{LN}^S, \mathcal{LN}^T)| \propto |Shift_{data}((X^S, Y^S), (X^T, Y^T))| \Rightarrow |Shift_{data}(X^S, X^T)|, \quad (2)$$

_where $Shift(\cdot)$ is an arbitrary quantification that measures the shifts between two variables._

_Proof._ Eq. (2) connects $\mathcal{LN}^S$ and $\mathcal{LN}^T$ via loss functions and assumptions, further detailed in Appendix A.4. □

---

[1]"_Overshoot_" occurs when the shift between (fine-tuned) $\mathcal{LN}^T$ and (original) $\mathcal{LN}^S$ surpasses the ideal shift $||\mathcal{LN}^T - \mathcal{LN}^S||_2 > ||\mathcal{LN} - \mathcal{LN}^S||_2$. "_Undershoot_" occurs when the shift is less than an ideal shift $||\mathcal{LN}^T - \mathcal{LN}^S||_2 < ||\mathcal{LN} - \mathcal{LN}^S||_2$.

The overall data distribution shift between source and target domains splits into input and label components: $|Shift_{data}((X^S, Y^S), (X^T, Y^T))| \Rightarrow |Shift_{data}(X^S, X^T)| + |Shift_{data}(Y^S, Y^T)|$. Here, $Shift_{data}(Y^S, Y^T)$ denotes label shift, often negligible in many real-world scenarios, as label distributions typically remain stable across domains for classification. Moreover, as in our approach Section 3, the $|Shift_{data}(Y^S, Y^T)|$ is captured by the firstly-tuned linear-based predictor and treated as constant, allowing us to focus on the $|Shift_{data}(X^S, X^T)|$, which is more likely to vary significantly and affect model adaptation. Thus, $Shift_{data}((X^S, Y^S), (X^T, Y^T)) \Rightarrow Shift_{data}(X^S, X^T)$.

Intuitively, ***while layers beyond LayerNorm are also capable of capturing distributional shifts, their learned representations are often entangled with semantic or task-specific factors, which can obscure or even degrade the ability of LayerNorm to isolate distributional shifts***. In contrast, LayerNorm primarily captures distributional variations in a more disentangled manner, focusing on normalization statistics without encoding additional task-specific knowledge. This property makes LayerNorm particularly suitable for isolating and adapting to domain shifts.

For $Shift_{ln}(\mathcal{LN}^T, \mathcal{LN}^S)$, it should reflect the global shifts of all LayerNorm layers: $Shift_{ln}(\mathcal{LN}^T, \mathcal{LN}^S) := \|\mathcal{LN}^T - \mathcal{LN}^S\|_{\mathcal{G}}$. Then, we can obtain:

$$\|\mathcal{LN}^T - \mathcal{LN}^S\|_{\mathcal{G}} := \sum\nolimits_{i=1}^n \left( \left\|\gamma_i^T - \gamma_i^S\right\|_2 + \left\|\beta_i^T - \beta_i^S\right\|_2 \right). \tag{3}$$

$Shift(X^S, X^T)$ can be denoted in various forms. Here, we adopt the Wasserstein distance to quantify $Shift(X^S, X^T)$ as it captures the overall geometric discrepancy between feature distributions rather than merely comparing their mean or variance: $Shift_{data}(X^S, X^T) := Wasserstein(X^S; X^T)$. This is particularly appropriate given that LayerNorm is sensitive to both mean and variance statistics, while the Wasserstein metric aligns naturally with its normalization objectives.

**Learned LayerNorm shifts versus ideal LayerNorm shifts.** We now study the gap among $\mathcal{M}^S$, $\mathcal{M}^T$, and $\mathcal{M}^{T*}$. According to Eq. (2), it links the shifts between LayerNorm and data distributions, if the distribution shifts between $X^S$, $X^T$, and $X^{T*}$ are characterized, the corresponding LayerNorm shifts can also be clarified. Here, $X^T$ is defined as the finite set of target-domain training samples used for fine-tuning, while $X^{T*}$ denotes the full target-domain data distribution (e.g., all available test samples). Since $X^T$ is typically a limited subset of $X^{T*}$, it may not fully capture the statistical properties of the full target domain. Thus, the distribution shift $shift_{data}(X^S, X^T)$ can differ from $shift_{data}(X^S, X^{T*})$. Consider the $shift_{data}(X^S, X^T)$ and $shift_{data}(X^S, X^{T*})$, the Fine-tuning Shift Ratio ($FSR$) is defined as

$$FSR := Shift_{data}(X^S, X^T)/(Shift_{data}(X^S, X^{T*}) + \epsilon), \tag{4}$$

where the $\epsilon$ is a small value to avoid numeric issues. Note that the shift is calculated in the raw space, and $X^T$ and $X^{T*}$ are from the same distribution, but. Following Eq. (2):

$$\frac{Shift_{ln}(\mathcal{LN}^T, \mathcal{LN}^S)}{Shift_{ln}(\mathcal{LN}^{T*}, \mathcal{LN}^S)} \propto FSR \Rightarrow \frac{Shift_{ln}(\mathcal{LN}^T, \mathcal{LN}^S)}{Shift_{ln}(\mathcal{LN}^{T*}, \mathcal{LN}^S)} = f(FSR), \tag{5}$$

where $f(\cdot)$ is a monotonic mapping function that captures the proportional relationship between $FSR$ and the distributional shifts. To prevent numeric issues, a small value like $1e-8$ will be added to the denominator of the omitted fractions. From Eq. (5), LayerNorm fine-tuning performance is influenced by the relative shift between target training samples and the source, rather than the absolute shift, compared to the full target domain and the source. It emphasizes that ***LayerNorm fine-tuning effectiveness is not inherently dependent on whether the target data is IP or ODD relative to the source.*** Rather, it depends on how well the target training data represents the overall target domain's distribution, especially regarding the source-target divergence. We further found that the target training data tend to be more representative in IP scenarios, but less so in OOP settings. See Section 4.2 for empirical analysis.

## 2.1 RESCALING LAYERNORM SHIFTS TO MIMIC FRS=1 SCENARIO

**Scenario one:** $FSR = 1$. $FSR = 1$ indicates the scenario where $X^T$ is statistical sufficient for representing $X^{T*}$. In such ideal cases, the fine-tuned LayerNorm layers can generalize for all $X^{T*}$.

**Scenario two:** $FSR < 1$ **or** $FSR > 1$. While $FSR$ dose not equals to one, the $X^T$ is no longer statistical sufficient for representing $X^{T*}$. Specifically, while $FSR > 1$, we have:

$$\frac{Shift_{ln}(\mathcal{LN}^T, \mathcal{LN}^S)}{Shift_{ln}(\mathcal{LN}^{T*}, \mathcal{LN}^S)} > f(FSR = 1) \Rightarrow \frac{\lambda \cdot Shift_{ln}(\mathcal{LN}^T, \mathcal{LN}^S)}{Shift_{ln}(\mathcal{LN}^{T*}, \mathcal{LN}^S)} = f(FSR = 1), \tag{6}$$

where $\lambda \geq 0$ and similar scenario for $FSR < 1$. Empirically, we notice that normally the $FSR \geq 1$ (see Section 4.1). Thus the tuned LayerNorm layers can be further improved with rescaling.

To investigate the rescale approach, we now consider only the $i^{th}$ layer of LayerNorm. Following Eq. (3), the left-hand sized of Eq. (6) can simplified as:

$$\frac{Shift_{ln}(\mathcal{LN}_i^T, \mathcal{LN}_i^S)}{Shift_{ln}(\mathcal{LN}_i^{T*}, \mathcal{LN}_i^S)} = \frac{\left\|\gamma_i^T - \gamma_i^S\right\|_2 + \left\|\beta_i^T - \beta_i^S\right\|_2}{\left\|\gamma_i^T - \gamma_i^S\right\|_2 + \left\|\beta_i^T - \beta_i^S\right\|_2}. \tag{7}$$

***We reveal that the convergence behaviors of the normalization parameters $\beta$ (mean) and $\gamma$ (variance) differ with respect to the number of target training samples $n$ if target training samples are randomly sampled.*** Due to the Central Limit Theorem, the sample mean (i.e., proxy for $\beta$) converges rapidly as $n$ increases, with estimation error decreasing at a rate of $\mathcal{O}(1/\sqrt{n})$. In contrast, variance estimation (proxy for $\gamma$) relies on the chi-squared distribution, converging slowly and requiring more samples for similar confidence. This asymmetry suggests that fewer samples are needed to adapt $\beta$ reliably than $\gamma$, aligning with our empirical findings in Table 5 and Appendix Fig. 11. Detailed derivations and sample complexity analysis of mean and variance estimation refer to Appendix A.4.

As analyzed above, the $\mu$ converge faster than $\sigma$ when given an increasing number of samples, indicating similar effects between $\beta$ and $\gamma$. Additionally, $\beta$ provides directional information, unlike $\sigma$. Since $\beta$ hasn't converged, its direction may also diverge, making rescaling it risky. Hence, only $\gamma^T$ should be rescaled. Lastly, $\lambda$ is applied to all $\beta^T$ for the following effect:

$$\frac{\lambda \cdot Shift_{ln}(\mathcal{LN}^T, \mathcal{LN}^S)}{Shift_{ln}(\mathcal{LN}^{T*}, \mathcal{LN}^S)} := \frac{\sum_i^n \left(\left\|\lambda \cdot \gamma_i^T - \gamma_i^S\right\|_2 + \left\|\beta_i^T - \beta_i^S\right\|_2\right)}{\sum_i^n \left(\left\|\gamma_i^{T*} - \gamma_i^S\right\|_2 + \left\|\beta_i^{T*} - \beta_i^S\right\|_2\right)} = f(FSR = 1). \tag{8}$$

All our experiments apply $\lambda$ to only $\gamma$ if there is no other specification.

# 3 BOOSTING LAYERNORM FINE-TUNING FOR REAL-WORLD APPLICATION

One presumption of when Eq. (2) holds is that the predictor and other parameters remain fixed. Most ViTFs lack a $\mathcal{C}$ for every downstream task, where the $\mathcal{C}$ needs to be constructed during the fine-tuning stage. Tuning LayerNorm layers with predictors, like LP-LN (Zhao et al., 2023), can destabilize target data distribution capture and degrade performance. Thus, our approach trains the predictors at first while the $\mathcal{M}$ is fixed (corresponding to Eq. (13) Data distribution gap term); and then tunes the LayerNorm layers while other parameters are fixed in a round (i.e., Eq. (13) Parameter adaptation gain term), as shown in Fig. 1. Such a round can be repetitive, lasting several turns to ensure the convergence of the overall model. Following (Kumar et al., 2022), we show this approach is feasible.

**Proposition 3.1.** *Let: $\mathcal{M}^S$ be a source model, and $\mathcal{C}^S$ be an MLP-based linear predictor (LP) paired with $\mathcal{M}^S$ such that the loss $L(\mathcal{C}^S, \mathcal{M}^S(X^S), Y^S)$ is minimized (where $L(\cdot)$ is the Mean Squared Error). Assume $\mathcal{C}^S$ is not available during fine-tuning. When freezing $\mathcal{M}^S$ (e.g., a frozen ViT backbone), there exists a target predictor $\mathcal{C}^{T'}$ such that $L(\mathcal{C}^S, \mathcal{M}^S(X^T), Y^T)$ is minimized. Then, the distance $d(\mathcal{C}^S, \mathcal{C}^{T'})$ is bounded.*

*Proof.* If $X^S$ and $X^T$ have no shift, Proposition 3.1 is obvious. We mainly discuss whether there is a shift between them. Considering the extracted features $\mathcal{Z}$ of $\mathcal{M}^S(X)$ and the raw data space $\mathcal{X}$. Since we have a well pretrained $\mathcal{M}^S$, following Proposition 3.1, we have $\cos \theta_{\max}(\mathcal{Z}, \mathcal{X}) > 0$. Denote parameters of $\mathcal{C}^{T'}$ as $v_{lp}^{T'}$, we have

$$L\left(v_{lp^{T'}}, Z^T\right)^{1/2} \leq (c/\cos\theta_{\max}(\mathcal{Z}, \mathcal{X}))^2 d\left(Z^T, Z^S\right) \|Z^{S^\top} v_{lp^S}\|_2^2, \tag{9}$$

where $c > 0$ is a constant; $d(\cdot, \cdot)$ is a distance measurement. Note Eq. (9) holds when the number of dimension of $v_{lp}$ when $i \to \infty$. Please refer to more details in Appendix A.4. $\square$

Thus, without fine-tuning LayerNorm, it is able to train $\mathcal{C}^{T'}$ with limited shifts from $\mathcal{C}^S$ using $X^T$. Then, we freeze the layers except the LayerNorm layers to minimize the Eq. (9), where the problem turns into $\min_{\mathcal{LN}^{T'}} L\left(\mathcal{LN}^{T'}, Z^T\right)$ which is equivalent to:

$$\min_{\mathcal{LN}^{T'}} \left(c/\cos\theta_{\max}(\mathcal{M}^{T'}(\mathcal{X}), \mathcal{X})\right)^2 d\left(Z^T, \mathcal{M}^{T'}(X^T)\right) \|\mathcal{M}^{T'}(X^T)^\top v_{lp}^{T'}\|_2^2, \tag{10}$$

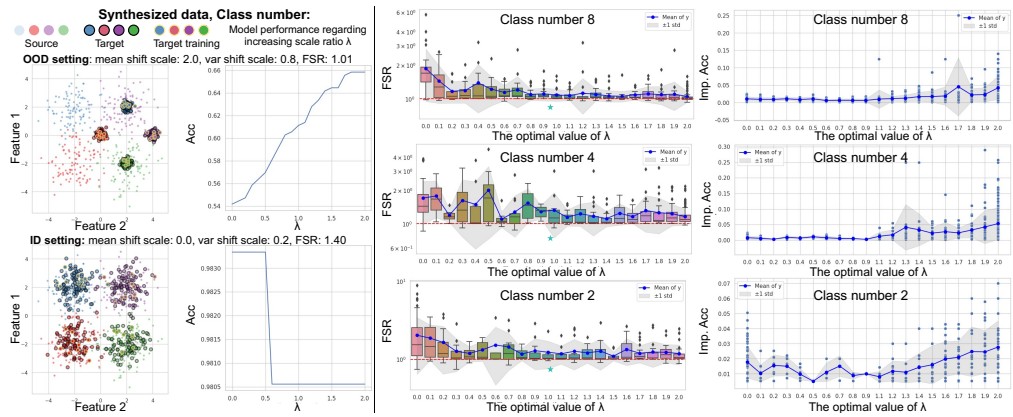

Figure 2: **Toy example**. Left: Two generated data examples and their rescaling performance. The target training samples are $10\%$ of all target samples. Right: Statistical visualization of $FSR$ and the optimal scaling factor $\lambda$ across different class numbers. Note: Models with zero accuracy or unchanged accuracy after rescaling are excluded. See code and results in the supplementary materials.

where $\mathcal{LN}^{T'}, \mathcal{M}^{T'}$ correspond to the LayerNorm layers and the overall model tuned with $\mathcal{C}^{T'}$. Clearly, tuning LayerNorm layers do not change the knowledge obtained by the overall model. The $\cos\theta_{\max}(\mathcal{M}^{T'}(\mathcal{X}), \mathcal{X})$ can be considered to be as very close to the $\cos\theta_{\max}(\mathcal{M}^{S}(\mathcal{X}), \mathcal{X})$, so does $\|\mathcal{M}^{T'}(X^T)^\top v_{\text{lp}}^{T'}\|_2^2$ to $\|Z^{S^T} v_{\text{lp}}^{T'}\|_2^2$. As discussed in Section 2, Layer-Norm fine-tuning captures the data distributional shifts, hence $d\left(Z^T, \mathcal{M}^{T'}(X^T)\right) \|\mathcal{M}^{T'}(X^T)\| \leq d\left(Z^T, Z^S\right) \|\mathcal{M}^{T'}(X^T)\|$. Therefore, the bounds of the loss become tighter than Eq. (10) during the the increasing number of turns. This indicates why the proposed method, as shown in Fig. 1, is feasible and stable (see Section 4.3). To further enhance improve LayerNorm fine-tuning, we introduce two additional techniques: (i) increasing feature dimensions before prediction, and (ii) applying lightweight feature augmentation before attention pooling. Details and ablations of them are in Appendix A.6.



Figure 1: Diagrams of the proposed training approach. One complete turn contains both rounds.

## 4 EXPERIMENTS

### 4.1 TOY EXAMPLES: THE RELATIONSHIP BETWEEN $\lambda$, $FSR$ AND PERFORMANCE

Computing the $FSR$ during fine-tuning is impractical due to the large scale and dataset accessibility. Pathological foundation models (Lu et al., 2024; Wang et al., 2024; Ding et al., 2024) use private datasets, complicating FSR measurement. We use controlled toy examples to test the proposed $\lambda$ and explore its connection with $FSR$.

**Toy example training and toy data generation.** We employ a 2-layer MLP with LayerNorm as a toy model to simulate fine-tuning in domain shift scenarios. Synthetic data, modeled as class-conditional Gaussians with mean and variance shifts, form source and target domains. We adjust the labeled target samples and shift levels to replicate various fine-tuning conditions. After LayerNorm fine-tuning, we rescale $\gamma$ by $\lambda \in [0, 2]$ to determine the optimal $\lambda$ for maximizing accuracy. See synthesized examples in Fig. 2 left-hand side and detailed descriptions in Appendix A.5.

Table 1: **Toy example**: Overall statistics of toy example results. Not affected cases: In the cases where the accuracy does not change with respect to scaling ratio values.

| | |
|---|---|
| Overall cases | 1815 |
| Not affected cases | 290 |
| Cases improved by $\lambda$ | 1326 |
| Averaged improvements of all improved cases | 2.39% |

Table 2: **IP and OOP**: Results of fine-tuning OpenCLIP and DINOv2 on SUN, DTD, and Bach.

| **OOP: OpenCLIP To SUN** | LP+WM | LP | LP-LN | Ours | Best λ value | +best λ | **OOP: OpenCLIP To DTD** | LP+WM | LP | LP-LN | Ours | Best λ value | +best λ |
|---|---|---|---|---|---|---|---|---|---|---|---|---|---|
| Fine-tuning data fraction: 2% | 34.78 | 20.97 | 32.27 | 35.49 | 1.0 | 35.49 | Fine-tuning data fraction: 2% | 0.00 | 0.00 | 0.00 | 6.38 | 1.2 | 6.38 |
| 5% | 58.24 | 51.29 | 59.35 | 58.70 | 1.5 | 59.06 | 5% | 32.98 | 11.70 | 37.23 | 45.74 | 1.2 | 46.81 |
| 10% | 64.30 | 61.96 | 67.30 | 66.13 | 1.1 | 66.15 | 10% | 50.53 | 25.00 | 47.34 | 51.06 | 1.2 | 51.06 |
| 100% | 73.68 | 76.13 | 75.19 | 76.70 | 1.3 | 76.80 | 100% | 48.83 | 79.47 | 79.20 | 79.89 | 1.2 | 80.05 |
| Avg | 57.75 | 52.59 | 58.53 | 59.25 | - | 59.37 | Avg | 33.09 | 29.04 | 40.94 | 45.77 | - | 46.08 |
| **OOP: OpenCLIP To Bach** | LP+WM | LP | LP-LN | Ours | Best λ value | +best λ | **OOP: DINOv2 To Bach** | LP+WM | LP | LP-LN | Ours | Best λ value | +best λ |
| Fine-tuning data Num. pre class: 1 | 33.50 | 44.50 | 31.50 | 37.00 | 1.2 | 37.50 | Fine-tuning data Num. pre class: 1 | 27.00 | 32.50 | 39.50 | 37.50 | 1.3 | 37.50 |
| 5 | 49.50 | 50.00 | 46.50 | 52.00 | 1.3 | 55.00 | 5 | 25.00 | 51.50 | 52.00 | 53.50 | 1.5 | 59.00 |
| 10 | 50.00 | 49.00 | 54.00 | 59.00 | 1.2 | 61.50 | 10 | 25.00 | 62.00 | 59.50 | 61.50 | 1.1 | 60.50 |
| 20 | 43.00 | 47.50 | 50.00 | 60.50 | 1.1 | 60.50 | 20 | 29.50 | 61.00 | 63.00 | 61.50 | 1.0 | 61.50 |
| All | 66.50 | 63.50 | 67.00 | 70.50 | 1.2 | 71.00 | All | 45.00 | 74.00 | 74.50 | 76.50 | 0.6 | 77.00 |
| Avg | 48.50 | 50.90 | 49.80 | 55.80 | - | 57.10 | Avg | 30.30 | 56.20 | 57.70 | 58.10 | - | 59.10 |
| **IP: DINOv2 To SUN** | LP+WM | LP | LP-LN | Ours | Best λ value | +best λ | **IP: DINOv2 To DTD** | LP+WM | LP | LP-LN | Ours | Best λ value | +best λ |
| Fine-tuning data fraction: 2% | 46.80 | 45.23 | 43.95 | 45.16 | 0.2 | 45.51 | Fine-tuning data fraction: 2% | 2.13 | 2.13 | 2.13 | 4.26 | 0.2 | 4.26 |
| 5% | 62.62 | 62.33 | 63.13 | 63.32 | 0.2 | 63.38 | 5% | 5.32 | 55.32 | 57.45 | 63.83 | 0.2 | 63.83 |
| 10% | 66.56 | 67.47 | 67.86 | 68.52 | 0.2 | 68.52 | 10% | 4.26 | 56.91 | 59.57 | 61.17 | 0.2 | 61.17 |
| 100% | 74.24 | 77.13 | 77.22 | 77.84 | 0.3 | 77.96 | 100% | 15.64 | 78.99 | 81.81 | 80.59 | 2.0 | 81.12 |
| Avg | 62.56 | 63.04 | 63.04 | 63.71 | - | 63.84 | Avg | 6.84 | 48.34 | 50.24 | 52.46 | - | 52.59 |

**Rescaling $\gamma$ with an appropriate $\lambda$ enhances target-domain performance.** As shown in Table 1, rescaling yields non-decreasing accuracy in over 90% of cases and leads to measurable gains in 73% of them, with an average improvement of 2.39%.

**The optimal $\lambda$ negatively correlates with $FSR$ yet positively with performance gains, suggesting that $\lambda$ can serve as an indicator for FSR when source data isn't available, but target data is.** As shown in Fig. 2 right-hand side, high $FSR$ values correspond to optimal $\lambda < 1$, while low $FSR$ values favor $\lambda > 1$ in most cases.

**The LayerNorm fine-tuning performance depends on how well the target training data represents the target domain under the influence of the source domain.** Fig. 2 left-hand side further illustrates this trend: in OOP settings with sufficient target data (lower $FSR$), $\lambda > 1$ leads to steady accuracy improvements; whereas in IP settings with limited data (larger $FSR$), $\lambda < 1$ results in smaller gains. This further supports that the performance of LayerNorm fine-tuning ties to how representative the target training data is to the target domain, rather than directly to IP or OOP settings.

Table 3: **OOP**: Results of domain generalization across unseen domains in the TerraIncognita (Beery et al., 2018) dataset using Vit-L14 (Dosovitskiy et al., 2021) pretrained on ImageNet.

| Vit-L14 | location_38 | location_46 | location_100 | location_43 |
|---|---|---|---|---|
| Only head | 67.505 | 62.779 | 68.785 | 70.403 |
| LP-LN | 68.160 | 61.547 | 73.214 | 71.127 |
| Ours | 68.558 | 61.887 | 74.479 | 70.057 |
| +λ=1.01 | 68.943 | 61.993 | 74.374 | 70.151 |

Table 4: **IP and OOP**: Results for segmentation adaptation with various methods (using officially released checkpoint) on Synthia (Ros et al., 2016) and CityScapes (Cordts et al., 2016) (CS), with CS as the target testing domain. Detailed results of each class are in Appendix Table 13.

| | mAcc |
|---|---|
| **IP**: Segformer (Xie et al., 2021) (CS->CS) | 96.73 |
| +λ=0.95 | **96.75** |
| **OOP**: Segformer+DAformer (Hoyer et al., 2022) (Synthia->CS) | 61.54 |
| +λ=1.05 | **61.64** |

**The observed tendencies are insensitive to the number of classes.** As shown in the right-hand side of Fig. 2, the same relationships between $FSR$, $\lambda$, and fine-tuning improvements are consistently observed across different numbers of classes. This suggests that neither $FSR$ nor the optimal $\lambda$ is significantly affected by the label space $Y$.

**Larger $FSR$ correlates with greater variance in the optimal value of $\lambda$, indicating increased randomness in LayerNorm convergence.** As shown in Fig. 2 right-hand side, smaller values of $FSR$ correspond to reduced variance in the optimal $\lambda$. This suggests that while $\lambda < 1$ tends to be beneficial for larger $FSR$, it is generally less effective compared to $\lambda > 1$ in cases with smaller $FSR$, highlighting the variability in LayerNorm fine-tuning effectiveness while $FSR$ is large.

## 4.2 REALISTIC DATASETS: $\lambda$ UNDER IP AND OOP SETTINGS

We observe that the optimal scaling parameter $\lambda$ exhibits a mild correlation with the IP or OOP setting, though it is not directly determined by the domain shift between source and target.

**Experimental settings.** We adopt natural image ViTFs including OpenCLIP (Radford et al., 2021) (ViT-B/32 pretrained on DataComp-1B), and DINOv2 (Oquab et al., 2023) (pretrained on LVD-142M) on natural image datasets, SUN (Xiao et al., 2010) and DTD (Cimpoi et al., 2014) (included by LVD-142M for DINOv2), accompanied by the pathological dataset, Bach (Aresta et al., 2019), for testing the IP and OOP settings for natural image ViTFs. Especially, different data regimes used for testing are selected from the testing domain, and the model is evaluated on held-out data. Evaluation results are taken from the final epoch, and $\lambda$ is further chosen based on test set performance as the indicator of $FSR$. See Appendix A.6.3 for full details.

Table 6: **Comparisons** (natural image results for OOP setting): Ours versus SOTA fine-tuning methods using MAE pre-trained on ImageNet Deng et al. (2009) on the DomainNet dataset domain splits: A = $\{Clipart, Infograph, Quickdraw\}$. B = $\{Real, Painting, Sketch\}$. Setting · **to** ·: pretrained on the former domain set, then fine-tuned and evaluated on the latter one. Setting **To ·, test on** ·: fine-tuned on the former domain set, then evaluated on the latter one.

| | | | | | | | | MAE | | | | | | | | |
|---|---|---|---|---|---|---|---|---|---|---|---|---|---|---|---|---|
| **To A, test on A** | LP | LP+FT | LoRA | LP-LN | Ours | Best λ | +best λ | | **To A, test on B** | LP | LP+FT | LoRA | LP-LN | Ours | Best λ | +best λ |
| Fine-tuning data fraction: 1% | 43.29 | 24.98 | 77.77 | 77.86 | 88.13 | 1.3 | 88.54 | | Fine-tuning data fraction: 1% | 39.80 | 24.03 | 43.58 | 45.71 | 61.29 | 2 | 62.16 |
| 5% | 59.56 | 49.83 | 92.20 | 93.24 | 91.38 | 1 | 91.41 | | 5% | 56.56 | 48.97 | 64.54 | 66.87 | 62.23 | 1.2 | 62.48 |
| 10% | 64.01 | 66.18 | 92.64 | 92.04 | 93.45 | 1.5 | 93.48 | | 10% | 58.80 | 58.06 | 65.59 | 64.05 | 67.29 | 1.5 | 67.77 |
| Avg. | 55.62 | 47.00 | 87.54 | 87.71 | 90.99 | - | 91.14 | | Avg. | 51.72 | 43.69 | 57.90 | 58.88 | 63.60 | - | 64.14 |
| **To B, test on B** | LP | LP+FT | LoRA | LP-LN | Ours | Best λ | +best λ | | **To B, test on A** | LP | LP+FT | LoRA | LP-LN | Ours | Best λ | +best λ |
| Fine-tuning data fraction: 1% | 45.48 | 23.34 | 43.70 | 45.00 | 70.00 | 1.3 | 70.24 | | Fine-tuning data fraction: 1% | 43.76 | 24.55 | 53.25 | 52.54 | 79.76 | 1.2 | 79.86 |
| 5% | 59.33 | 47.47 | 79.02 | 81.34 | 77.51 | 0.8 | 77.51 | | 5% | 59.22 | 24.55 | 85.20 | 86.65 | 78.98 | 1.1 | 78.98 |
| 10% | 64.81 | 60.65 | 79.78 | 80.53 | 82.79 | 1.2 | 82.86 | | 10% | 64.00 | 65.21 | 87.57 | 83.46 | 89.61 | 1.5 | 89.67 |
| Avg. | 56.54 | 43.82 | 67.50 | 68.96 | 76.77 | - | 76.87 | | Avg. | 55.66 | 38.10 | 75.34 | 74.22 | 82.78 | - | 82.84 |

**Comparisons.** For comparison, we additionally conduct (1) LP+WM: Fine-tuning the whole $\mathcal{M}$ with LP, (2) LP: Training the LP model with the frozen $\mathcal{M}$, (3) LP-LN (Zhao et al., 2023): Current State-of-the-art (SoTA) LayerNorm fine-tuning which tunes the $\mathcal{LN}$ with LP simultaneously. and (4) Ours: our proposed method that is introduced in Section 3. All experiments, including our method and the comparisons, use the same overall training epochs and learning rates for a fair comparison.

**Results.** Table 2 shows that our method outperforms other competitors, where more experiments regarding our method can be seen in Section 4.3. Specifically, Fig. 3 shows that $\lambda < 0$ for IP and $\lambda > 0$ for OOP settings under most scenarios. This suggests that in OOP scenarios, the limited target training data is more likely to be insufficient to fully represent the target domain distribution, leading to a larger $FSR$. *Therefore, while the setting is explicit, we recommend setting $\lambda$ slightly larger than 1 and smaller than 1 for OOP and IP settings, respectively.* Moreover, as the relationship between $\lambda$ and IP/OOP settings that we have revealed, it is possible to use the optimal value of $\lambda$ to gain insight into the current fine-tuning scenario when the source data distribution is intractable but the testing domain is available.

**Extended results for multi-domain generalization and semantic segmentation tasks.** As Tables 3 and 4 we further tested $\lambda$ with our method on the domain generalization task and segmentation adaptation task for the known OOP (using $\lambda > 1$) and IP (using $\lambda < 1$) settings. It can be seen that using proper $\lambda$ can still lead to improvements in most cases (classes or domains), suggesting that our scaling concept can be applied to generalization or adaptation settings for broader tasks.

**Convergence of $\beta$ and $\gamma$.** As detailed in Section 2.1, Table 5 shows that $\beta$ consistently converges faster than $\gamma$ as the number of fine-tuning samples increases. Notably, rescaling $\gamma$ while $\beta$ remains unconverged may fail to improve model performance; further analysis and supporting evidence are provided in Section 4.4.

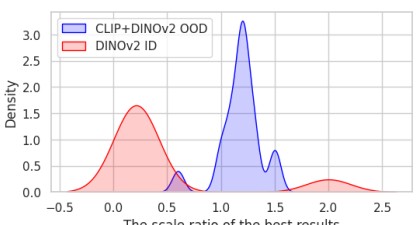

Figure 3: **IP and OOP**: Kernel density estimation visualization of the optimal (best) values of $\lambda$ from Table 2.

Table 5: **Convergence** across data regimes (Conch (Lu et al., 2024) on Bach (Aresta et al., 2019) dataset). Convergence is evaluated as normed tuned $\gamma$, $\beta$ compared with normed best $\gamma^*$, $\beta^*$.

| Num. of samples pre class | mse($\gamma,\gamma^*$) | mse($\beta,\beta^*$) |
|---|---|---|
| 1 | 1.31e-04 | 0.079153 |
| 5 | 9e-05 | 0.079208 |
| 10 | 9.5e-05 | 0.079075 |
| 20 | 0.0 | 0.079033 |

### 4.3 MORE EXPERIMENTS OF OUR METHOD

As the proposed method has been validated in Section 4.2, we conduct further comparison to more other methods.

**Compared to fine-tuning strategies without extra parameters.** We compare our proposed fine-tuning strategy against those without extra parameters, where our approach yields the best results. Specifically, (Kumar et al., 2022) proposes an LP-FT strategy, where the linear-probability (LP) predictor is fine-tuned first, followed by fine-tuning of the entire model, including all layers in $\mathcal{M}$ and the predictor. However, our experiments, as shown in

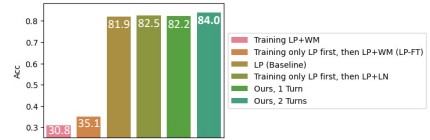

Figure 4: **Comparisons**: Ours versus different fine-tuning strategies without extra parameters. Results show average performance across varying fine-tuning data fractions on the Bach dataset using the CONCH. WM: whole model $\mathcal{M}$; see detailed results in Appendix A.6.

Table 7: **Natural image results**: Ours versus other fine-tuning methods using CLIP and DINOv2 on the DomainNet domain splits.

**CLIP**

| To A, test on A | LP+WM | LP | LP-LN | Ours | Best λ | +best λ | +λ=1.2 | To A, test on B | LP+WM | LP | LP-LN | Ours | Best λ | +best λ | +λ=1.2 |
|---|---|---|---|---|---|---|---|---|---|---|---|---|---|---|---|
| Fine-tuning data fraction: 1% | 36.98 | 75.45 | 77.86 | 88.13 | 2 | 88.54 | 88.28 | Fine-tuning data fraction: 1% | 17.84 | 42.72 | 45.71 | 61.29 | 2 | 62.16 | 61.36 |
| 5% | 62.74 | 92.31 | 93.24 | 91.38 | 1.3 | 91.41 | 91.38 | 5% | 32.07 | 64.68 | 66.87 | 62.23 | 1.3 | 62.48 | 62.41 |
| 10% | 83.35 | 50.41 | 57.29 | 93.45 | 1.5 | 93.48 | 93.5 | 10% | 54.85 | 28.29 | 29.7 | 67.29 | 1.5 | 67.77 | 67.53 |
| 50% | 86.78 | 90.49 | 92.25 | 94.44 | 1.1 | 94.45 | 94.41 | 50% | 57.37 | 62.5 | 64.61 | 67.02 | 1.1 | 67.07 | 67.25 |
| Avg | 67.46 | 77.17 | 80.16 | 91.85 | - | 91.97 | 91.89 | Avg | 40.53 | 49.55 | 51.72 | 64.46 | - | 64.87 | 64.64 |

| To B, test on B | LP+WM | LP | LP-LN | Ours | Best λ | +best λ | +λ=1.2 | To B, test on A | LP+WM | LP | LP-LN | Ours | Best λ | +best λ | +λ=1.2 |
|---|---|---|---|---|---|---|---|---|---|---|---|---|---|---|---|
| Fine-tuning data fraction: 1% | 13.11 | 43.15 | 45 | 70 | 1.2 | 70.24 | 70.24 | Fine-tuning data fraction: 1% | 9.98 | 54.48 | 52.54 | 79.76 | 1.2 | 79.86 | 79.86 |
| 5% | 56.13 | 79.07 | 81.34 | 77.51 | 1 | 77.51 | 77.26 | 5% | 41.71 | 84.64 | 86.65 | 78.98 | 1 | 78.98 | 79.02 |
| 10% | 71.52 | 33.08 | 32.36 | 82.79 | 1.2 | 82.86 | 82.86 | 10% | 67.3 | 39 | 36.66 | 89.61 | 1.2 | 89.67 | 89.67 |
| 50% | 75.98 | 76.28 | 79.02 | 86.55 | 1.2 | 86.61 | 86.61 | 50% | 67.5 | 85.81 | 86.81 | 90.18 | 1.2 | 90.11 | 90.11 |
| Avg | 54.19 | 57.9 | 59.43 | 79.21 | - | 79.31 | 79.24 | Avg | 46.62 | 65.98 | 65.67 | 84.63 | - | 84.66 | 84.67 |

**DINOv2**

| To A, test on A | LP+WM | LP | LP-LN | Ours | Best λ | +best λ | +λ=1.2 | To A, test on B | LP+WM | LP | LP-LN | Ours | Best λ | +best λ | +λ=1.2 |
|---|---|---|---|---|---|---|---|---|---|---|---|---|---|---|---|
| Fine-tuning data fraction: 1% | 72.76 | 80.33 | 83.16 | 90.29 | 1.3 | 90.37 | 90.36 | Fine-tuning data fraction: 1% | 34.13 | 43.17 | 48.86 | 55.78 | 1.3 | 56.1 | 55.97 |
| 5% | 90.21 | 92.34 | 93.71 | 93.48 | 1.3 | 93.56 | 93.5 | 5% | 46.41 | 57.81 | 62.98 | 62.81 | 1.3 | 63.16 | 62.99 |
| 10% | 90.21 | 73.75 | 84.19 | 93.74 | 1.3 | 93.82 | 93.75 | 10% | 57.28 | 41.31 | 48.65 | 62.47 | 1.3 | 62.62 | 62.61 |
| 50% | 93.79 | 92.44 | 94.71 | 95.79 | 1.3 | 95.97 | 95.92 | 50% | 62.49 | 58.69 | 65.08 | 66.4 | 1.3 | 66.76 | 66.63 |
| Avg | 86.74 | 84.72 | 88.94 | 93.33 | - | 93.43 | 93.38 | Avg | 50.08 | 50.25 | 56.39 | 61.87 | - | 62.16 | 62.05 |

| To B, test on B | LP+WM | LP | LP-LN | Ours | Best λ | +best λ | +λ=1.2 | To B, test on A | LP+WM | LP | LP-LN | Ours | Best λ | +best λ | +λ=1.2 |
|---|---|---|---|---|---|---|---|---|---|---|---|---|---|---|---|
| Fine-tuning data fraction: 1% | 50.71 | 50.71 | 55.16 | 70.06 | 1.5 | 70.82 | 70.44 | Fine-tuning data fraction: 1% | 62.17 | 48.03 | 62.5 | 80.14 | 1.5 | 80.45 | 80.34 |
| 5% | 79.9 | 79.9 | 82.55 | 81.21 | 1.1 | 81.13 | 81.12 | 5% | 83.71 | 70.14 | 87.11 | 86.3 | 1.1 | 86.4 | 86.47 |
| 10% | 47.47 | 47.47 | 66.05 | 81.53 | 1.3 | 81.75 | 81.68 | 10% | 49.53 | 83.46 | 68.03 | 90.12 | 1.3 | 90.31 | 90.29 |
| 50% | 79.48 | 79.48 | 84.76 | 87.37 | 1.2 | 87.44 | 87.44 | 50% | 85.51 | 83.67 | 90.33 | 91.25 | 1.2 | 91.28 | 91.28 |
| Avg | 64.39 | 64.39 | 72.13 | 80.04 | - | 80.29 | 80.17 | Avg | 70.23 | 71.33 | 76.99 | 86.95 | - | 87.11 | 87.1 |

Table 8: **Pathological image results for OOP settings**: Ours versus other fine-tuning methods using CHEIF, CONCH, and TITAN on the four pathological datasets. Please refer to more details for each model in Appendix Fig. 13.

**BACH**

| CHEIF | LP | LP-LN | Ours | Best λ | Ours + best λ | CONCH | LP | LP-LN | Ours | Best λ | Ours + best λ | TITAN | LP | LP-LN | Ours | Best λ | Ours + best λ |
|---|---|---|---|---|---|---|---|---|---|---|---|---|---|---|---|---|---|
| 1 | 0.535 | 0.520 | 0.550 | 1 | 0.555 | 1 | 0.685 | 0.635 | 0.675 | 0.1 | 0.700 | 1 | 0.655 | 0.610 | 0.665 | 0.1 | 0.685 |
| 5 | 0.615 | 0.690 | 0.700 | 1 | 0.700 | 5 | 0.825 | 0.825 | 0.795 | 0.3 | 0.805 | 5 | 0.840 | 0.805 | 0.840 | 1 | 0.840 |
| 10 | 0.720 | 0.745 | 0.755 | 1.5 | 0.755 | 10 | 0.815 | 0.840 | 0.860 | 2 | 0.870 | 10 | 0.825 | 0.815 | 0.845 | 2 | 0.850 |
| 20 | 0.790 | 0.785 | 0.810 | 2 | 0.835 | 20 | 0.865 | 0.865 | 0.870 | 2 | 0.890 | 20 | 0.850 | 0.880 | 0.915 | 2 | 0.925 |
| All | 0.855 | 0.890 | 0.870 | 2 | 0.875 | All | 0.905 | 0.935 | 0.930 | 2 | 0.950 | All | 0.950 | 0.935 | 0.965 | 1.3 | 0.970 |
| Avg. | 0.703 | 0.726 | 0.737 | - | **0.744** | Avg. | 0.819 | 0.820 | 0.826 | - | **0.843** | Avg. | 0.824 | 0.809 | 0.846 | - | **0.854** |

**Breakhis**

| CHEIF | LP | LP-LN | Ours | Best λ | Ours + best λ | CONCH | LP | LP-LN | Ours | Best λ | Ours + best λ | TITAN | LP | LP-LN | Ours | Best λ | Ours + best λ |
|---|---|---|---|---|---|---|---|---|---|---|---|---|---|---|---|---|---|
| 1 | 0.435 | 0.480 | 0.484 | 2 | 0.494 | 1 | 0.436 | 0.446 | 0.490 | 1 | 0.490 | 1 | 0.416 | 0.410 | 0.486 | 1 | 0.486 |
| 5 | 0.405 | 0.290 | 0.405 | 1 | 0.405 | 5 | 0.407 | 0.465 | 0.405 | 1 | 0.405 | 5 | 0.405 | 0.365 | 0.411 | 1 | 0.411 |
| 10 | 0.502 | 0.599 | 0.587 | 2 | 0.592 | 10 | 0.445 | 0.518 | 0.524 | 2 | 0.537 | 10 | 0.469 | 0.539 | 0.547 | 1.3 | 0.552 |
| All | 0.589 | 0.670 | 0.663 | 1.2 | 0.663 | All | 0.486 | 0.577 | 0.553 | 2 | 0.556 | All | 0.538 | 0.588 | 0.589 | 2 | 0.596 |
| Avg. | 0.483 | 0.510 | 0.535 | - | **0.539** | Avg. | 0.443 | **0.502** | 0.493 | - | 0.497 | Avg. | 0.457 | 0.475 | 0.508 | - | **0.511** |

**CCRCC Binary**

| CHEIF | LP | LP-LN | Ours | Best λ | Ours + best λ | CONCH | LP | LP-LN | Ours | Best λ | Ours + best λ | TITAN | LP | LP-LN | Ours | Best λ | Ours + best λ |
|---|---|---|---|---|---|---|---|---|---|---|---|---|---|---|---|---|---|
| 1 | 0.801 | 0.801 | 0.801 | 1 | 0.801 | 1 | 0.801 | 0.801 | 0.801 | 1 | 0.801 | 1 | 0.801 | 0.801 | 0.801 | 1 | 0.801 |
| 5 | 0.801 | 0.796 | 0.801 | 1 | 0.801 | 5 | 0.801 | 0.626 | 0.801 | 1 | 0.801 | 5 | 0.801 | 0.810 | 0.801 | 1 | 0.801 |
| 10 | 0.801 | 0.799 | 0.801 | 1 | 0.801 | 10 | 0.801 | 0.659 | 0.801 | 1 | 0.801 | 10 | 0.801 | 0.834 | 0.801 | 1 | 0.801 |
| 20 | 0.875 | 0.885 | 0.877 | 1.2 | 0.878 | 20 | 0.801 | 0.871 | 0.879 | 1.2 | 0.879 | 20 | 0.801 | 0.845 | 0.831 | 1.5 | 0.846 |
| Avg. | 0.819 | 0.820 | 0.820 | - | **0.820** | Avg. | 0.801 | 0.739 | 0.820 | - | **0.820** | Avg. | 0.801 | **0.823** | 0.808 | - | 0.812 |

**CCRCC Tissue**

| CHEIF | LP | LP-LN | Ours | Best λ | Ours + best λ | CONCH | LP | LP-LN | Ours | Best λ | Ours + best λ | TITAN | LP | LP-LN | Ours | Best λ | Ours + best λ |
|---|---|---|---|---|---|---|---|---|---|---|---|---|---|---|---|---|---|
| 1 | 0.672 | 0.713 | 0.667 | 0.1 | 0.673 | 1 | 0.699 | 0.668 | 0.715 | 2 | 0.719 | 1 | 0.720 | 0.695 | 0.707 | 1.5 | 0.708 |
| 5 | 0.759 | 0.743 | 0.735 | 0.7 | 0.743 | 5 | 0.797 | 0.769 | 0.825 | 1 | 0.825 | 5 | 0.803 | 0.677 | 0.829 | 2 | 0.840 |
| 10 | 0.769 | 0.778 | 0.764 | 2 | 0.764 | 10 | 0.668 | 0.787 | 0.816 | 1 | 0.816 | 10 | 0.712 | 0.805 | 0.828 | 2 | 0.832 |
| 20 | 0.811 | 0.826 | 0.807 | 2 | 0.807 | 20 | 0.784 | 0.836 | 0.841 | 1 | 0.841 | 20 | 0.798 | 0.822 | 0.851 | 2 | 0.852 |
| Avg. | 0.752 | **0.765** | 0.743 | - | 0.747 | Avg. | 0.737 | 0.765 | 0.799 | - | **0.800** | Avg. | 0.759 | 0.750 | 0.804 | - | **0.808** |

Fig. 4, indicate that this approach is generally ineffective for most ViTF fine-tuning scenarios. Given the typically limited amount of target training data, fine-tuning the full model, even with the LP-FT strategy, often leads to notable performance degradation. Moreover, when the LP-FT strategy is applied with fine-tuning restricted to LayerNorm, performance still deteriorates due to a violation of the $\mathcal{C}$-fixing presumption, as discussed in Section 2. Moreover, we provide boxplots of our proposed method across various seeds against others in Appendix Fig. 15, where stable improvements with relatively smaller variances indicate the robustness of our proposed method.

**Compared to SOTA fine-tuning methods.** We further compare our proposed method to SOTA methods Table 6 reports results for LP-FT (Kumar et al., 2022), LP-LN (Zhao et al., 2023), and LoRA (Hu et al., 2022), with LP included as additional baselines. Our method consistently achieves the best performance across different data regimes, with particularly strong gains under the 1% fine-tuning setting. Moreover, the optimal $\lambda$ identified in our approach suggests further improvements can be obtained through rescaling. Additional results across multiple seeds are provided in Appendix Table 14, where our method shows better robustness as it achieves the highest average performance with comparatively lower variance across various data regimes.

## 4.4 FURTHER EXPLORATION FOR OOP SETTINGS

Due to most current fine-tuning scenarios falling in OOP settings, we further adopt natural images and pathological ViTFs for their corresponding fine-tuning using different fine-tuning data regimes.

**Experimental settings: Fine-tuning natural images ViTFs on natural images.** We fine-tune the most commonly used natural image ViTFs on DomainNet domain splits, including CLIP (ViT-B/32) (Radford et al., 2021), OpenCLIP (ViT-B/32 pretrained on DataComp-1B), and DINOv2

(vitb14_reg pretrained on LVD-142M) (Oquab et al., 2023), with other settings as in Section 4.2. **Fine-tuning pathological ViTFs on pathological images.** We evaluate the fine-tuning performance of pathological ViTFs (CONCH (Lu et al., 2024), CHEIF (Wang et al., 2024), and TITAN (Ding et al., 2024)) using five benchmark datasets: Bach (Aresta et al., 2019), Breakhis (Spanhol et al., 2015), CCRCC (Brummer et al., 2023) (including both binary and tissue subsets), and Glas (Sirinukunwattana et al., 2017). Similar to natural image ViTFs, $\lambda$ is further chosen based on test set performance. More experimental details and results are in Appendix A.7.

**Results.** Our experimental results on natural images (Tables 6 and 7) and pathological images (Table 2) show that the proposed method outperforms standard LayerNorm fine-tuning across most scenarios. Notably, we mark in orange the cases where the optimal $\lambda$ is less than 1. In general, OOP settings yield optimal $\lambda$ values greater than 1, though this trend breaks down when fine-tuning data is extremely scarce. For example, as shown in Table 2, the exception occurs when fewer than five examples per class are available for fine-tuning. This observation aligns with our analysis in Section 2.1, which indicates that LayerNorm fine-tuning convergence depends on sample size: with too few samples, $\beta$ may not converge, and scaling $\gamma$ downward fails to improve performance. Additional results and detailed analysis for natural and pathological domains are provided in Appendices A.6.3 and A.7.

**Other findings in LayerNorm fine-tuning.** Figure 5 visualizes the shifts in all LayerNorm parameters, the scale $\gamma$ and bias $\beta$ within the $\mathcal{M}$, where even-numbered layers correspond to $ln^1$ and odd-numbered layers to $ln^2$. The visualization reveals several key insights. (1) In our cases, $ln^1$ exhibits greater shifts and thus contributes more significantly to fine-tuning performance than its $ln^2$ counterparts. (2) The specific LayerNorm layers requiring tuning vary across different ViT architectures, indicating model-dependent behavior rather than a consistent relationship with the number of target training samples. Thus, tuning all LayerNorm layers would be a robust approach for all ViTFs. (3) Shifts in $\gamma$ and $\beta$ are not always synchronized across layers (e.g., for CONCH), suggesting that scale and bias may play distinct roles during adaptation. (4) Increasing the number of tuning samples generally leads to sparser LayerNorm updates and

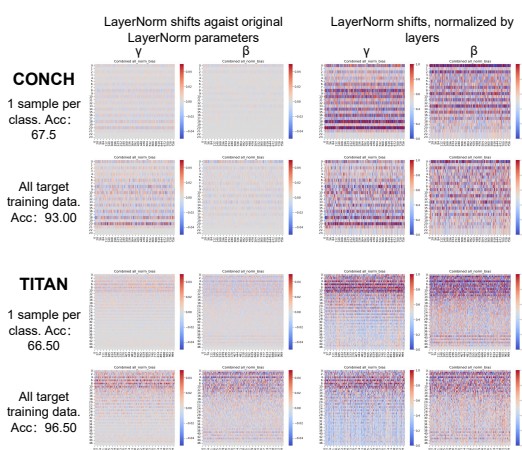

Figure 5: **Other findings**: Visualizations of LayerNorm shifts on the Bach dataset. Details in Appendix Fig. 12.

better performance, implying greater confidence in fewer parameters. Sparsifying LayerNorm updates in low-data regimes is beneficial but challenging due to model-specific complexities and sample selection uncertainties. More attempts and the corresponding results can be seen in Appendix A.7.3.

## 5 CONCLUSION AND DISCUSSIONS

**Discussions.** This paper investigates the interaction between LayerNorm fine-tuning and varying amounts of target training data with domain shifts. We find that LayerNorm fine-tuning tends to overshoot in IP tasks, whereas it typically undershoots in OOP tasks when sufficient fine-tuning samples are available. These effects can be mitigated through LayerNorm parameter rescaling with a scalar $\lambda$ and further improved via our proposed fine-tuning approach. Moreover, optimal $\lambda$ values may serve as an indicator of target data sufficiency, particularly when the ID/OOD status is uncertain, guiding future studies. **Future work.** Currently, our study focuses on the visual domain, where domain shifts are more easily quantified and interpreted. We hypothesize that similar behaviors may arise in language transformers, which is left for future work. Finally, while our analysis concentrates on standard LayerNorm, alternative normalization variants (Xu et al., 2019) may exhibit analogous patterns, and architectures such as DiT (Peebles & Xie, 2023) or transformers without explicit normalization layers (Zhu et al., 2025) are beyond the scope of this study. **Limitations.** Our study only focuses on classification with linear predictors to isolate LayerNorm effects; extensions to more complex tasks and models are discussed in Appendix A.2.

## ETHICS STATEMENT

Our study does not involve human subjects, sensitive data, or applications that may pose ethical risks. We believe this work raises no ethical concerns.

## REPRODUCIBILITY STATEMENT

The methodology is detailed in both the main paper and the appendices. Source code for key experiments is provided in the supplementary material and will also be publicly released, ensuring that all experimental results can be fully reproduced.

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

# A APPENDIX

## A.1 USAGE OF LARGE LANGUAGE MODELS

We used Large Language Models (LLMs) only as writing aids, for grammar checking and minor stylistic improvements. No LLMs were involved in designing experiments, analyzing data, or contributing to the scientific findings of this work.

## A.2 LIMITATIONS

Our study is limited to classification tasks, where the predictor $\mathcal{C}$ is linear, in order to isolate the effects of LayerNorm fine-tuning and avoid confounding factors introduced by more complex predictors. Extending our framework to other tasks, such as detection, segmentation, or multi-task learning, remains an important direction for future work. In multi-task scenarios, it is plausible that different LayerNorm scale parameters ($\gamma$) may require distinct rescaling directions, i.e., some increasing, others decreasing, depending on the specific sub-task characteristics. Moreover, our approach explicitly

assumes that the source model $\mathcal{M}^S$ is well-trained and capable of extracting all necessary information from $X$ for predicting $Y$. If the target task involves novel label spaces or requires information not captured by $\mathcal{M}^S$, fine-tuning only the LayerNorm layers would likely be insufficient. Nevertheless, in such cases, combining LayerNorm tuning with additional mechanisms that enrich the model's feature representations could further enhance downstream performance.

### A.3 RELATED WORK

**Studies of normalizations.** Normalization layers have become integral components of deep learning architectures across both language and vision domains. The seminal work on Batch Normalization (Ioffe & Szegedy, 2015) introduced normalization as a means to mitigate internal covariate shift, thereby improving training stability and convergence. Since then, various normalization techniques have been proposed, including Instance Normalization (Ulyanov et al., 2016), Group Normalization (Wu & He, 2018), and Layer Normalization (Ba et al., 2016). Among these, LayerNorm has emerged as the standard choice in the design of large-scale models, particularly in large language models (LLMs) and visual transformer foundation models (ViTFs). In this work, we focus specifically on the role of LayerNorm within ViTFs.

**Parameter efficient transfer learning.** Parameter-Efficient Transfer Learning. Our work is closely related to parameter-efficient transfer learning, within which LayerNorm fine-tuning represents a specific subcase. Broadly, parameter-efficient methods either introduce additional lightweight trainable components, such as adapters (Houlsby et al., 2019) or prompt tokens (Jia et al., 2022), into a frozen ViTF, or selectively fine-tune a subset of the model's existing parameters, such as all bias terms (Zaken et al., 2021). More recently, LoRA (Hu et al., 2022) has demonstrated that optimizing low-rank decomposition matrices for weight updates in dense layers offers an effective and scalable approach to adapting large models.

**LayerNorm fine-tuning approaches.** For LLMs, (Zhao et al., 2023) has demonstrated that Layer-Norm fine-tuning is both effective and sufficient for multi-modal LLMs, outperforming widely used methods such as LoRA (Hu et al., 2022), a finding further supported by (Chen et al., 2024). Similarly, (ValizadehAslani & Liang, 2024) emphasizes the centrality of LayerNorm in parameter-efficient fine-tuning strategies. In the context of ViTFs, (De Min et al., 2023) shows that LayerNorm tuning is likewise efficient and proposes introducing auxiliary learnable parameters to capture LayerNorm shifts instead of modifying the original parameters directly. Despite these promising results, existing studies have not sufficiently examined the underlying mechanisms of LayerNorm during fine-tuning, nor have they explored how its effectiveness might be further enhanced.

**Visual transformer foundation models.** Visual transformer foundation models (ViTFs) can broadly be categorized into two main branches. The first focuses on aligning multi-modal representations, particularly between text and images, as exemplified by models such as CLIP and OpenCLIP (Radford et al., 2021). These multi-modal ViTFs have extended their influence beyond natural images, gaining traction in medical imaging domains, like pathological images, through models like CONCH (Lu et al., 2024), CHEIF (Wang et al., 2024), and TITAN (Ding et al., 2024). The second branch comprises models trained exclusively on visual modalities, such as DINOv2 (Oquab et al., 2023), which leverage self-supervised learning frameworks. In particular, methods like Masked Autoencoders (MAE) (He et al., 2022) have demonstrated strong potential in advancing the training of ViTFs within this branch.

### A.4 MATHEMATICAL DETAILS

**Proof details of Proposition 2.1** To achieve that, we connect $\mathcal{LN}^S$ and $\mathcal{LN}^T$ by introducing definitions and assumptions on the loss function. The overall loss of a model on a given dataset $(X, Y)$ can be defined as:

$$L(\mathcal{C} \circ \mathcal{M}, X, Y) = \mathbb{E}_{(x,y)\sim(X,Y)}[\ell(\mathcal{C}(\mathcal{M}(x)), y)]. \tag{11}$$

Now we separating LayerNorm parameters $\mathcal{LN}$ from other parameters and $\mathcal{C}$ denotes the predictor, we have:

$$L(\mathcal{C} \circ \mathcal{M}, X, Y) = \mathbb{E}_{(x,y)\sim(X,Y)}[\ell(\mathcal{C}(\mathcal{M}_{\mathcal{LN}}(x)), y)] + \mathbb{E}_{(x,y)\sim(X,Y)}[\ell(\mathcal{C}(\mathcal{M}_{/\mathcal{LN}}(x)), y)], \tag{12}$$

where $\mathcal{M}_{\mathcal{LN}}$ is the $\mathcal{LN}$ in the model without and $\mathcal{M}_{/\mathcal{LN}}$ is the rest of the model without $\mathcal{LN}$.

Given an original model trained on the source dataset $(X^S, Y^S)$ and a fine-tuned model on the target dataset $(X^T, Y^T)$, we compare their losses to characterize the relationship between the source-trained model $\mathcal{C}^S \circ \mathcal{M}^S$ and the target-trained model $\mathcal{C}^T \circ \mathcal{M}^T$. Specifically, we define

$$\Delta L := L(\mathcal{C}^S \circ \mathcal{M}^S, X^S, Y^S) - L(\mathcal{C}^T \circ \mathcal{M}^T, X^T, Y^T) = \Delta L_{\mathcal{C}} + \Delta L_{\mathcal{M}_{/\mathcal{LN}}} + \Delta L_{\mathcal{LN}},$$

where $\Delta L_{\mathcal{C}}, \Delta L_{\mathcal{M}_{/\mathcal{LN}}}$, and $\Delta L_{\mathcal{LN}}$ denote the respective contributions of the classifier $\mathcal{C}$, the backbone excluding normalization layers $\mathcal{M}_{/\mathcal{LN}}$, and the normalization layers $\mathcal{LN}$.

Specifically, as the gap between $Y^T, Y^S$ is assumed relatively stable in compared with $X^T, X^S$, we ignore the shifts in $Y$. Moreover, in our approach, we fine-tune the linear-based $\mathcal{C}$ at first and then tune the $\mathcal{LN}$, which bridges the gap between $Y^T, Y^S$. Thus, if other parameters except the $\mathcal{LN}$ are fixed and $\Delta L_{\mathcal{C}} = 0, \Delta L_{\mathcal{M}_{/\mathcal{LN}}} = 0$, we have that

$$\Delta L = \underbrace{L\left(\mathcal{LN}^S, X^S\right) - L\left(\mathcal{LN}^S, X^T\right)}_{\text{Domain gap loss}} + \underbrace{L\left(\mathcal{LN}^S, X^T\right) - L\left(\mathcal{LN}^T, X^T\right)}_{\text{Parameter adaptation gain}}. \tag{13}$$

Assuming both models are well-optimized on their respective datasets, i.e., $\Delta L \approx 0$ since $L(\mathcal{LN}^S, X^S, Y^S) \approx 0, L(\mathcal{LN}^T, X^T, Y^T) \approx 0$. This implies that:

$$L\left(\mathcal{LN}^S, X^T\right) - L\left(\mathcal{LN}^T, X^T\right) \approx L\left(\mathcal{LN}^S, X^T\right) - L\left(\mathcal{LN}^S, X^S\right). \tag{14}$$

That is, the performance gain from adapting $\mathcal{LN}$ to the target domain is roughly equal to the distribution shift between source and target data **under other parameters in the $\mathcal{M}$ and $\mathcal{C}$ are fixed**. Thus, we have Proposition 2.1.

However, if $\Delta L_{\mathcal{C}} \neq 0$ or $\Delta L_{\mathcal{M}_{/\mathcal{LN}}} \neq 0$ where other parameters beside LayerNorm are not fixed, then Proposition 2.1 may not hold. Specifically, training $\Delta L_{\mathcal{C}}$ with $\mathcal{LN}$ may actually degrade the performance of the model. Please refer to the results in Fig. 4.

**Details analysis of Section 2.1.** One crucial problem here is that the convergence behavior of the normalization parameters $\gamma$ and $\beta$ differs depending on the number of target training samples $n$. Here, we assume that all target training samples for fine-tuning are randomly sampled from the target domain.

In Layer Normalization, $\beta$ and $\gamma$ are learnable affine parameters applied after normalization. During adaptation or estimation under domain shift, they are often initialized or updated based on batch statistics: $\beta$ can be associated with the estimated mean $\mu$, while $\gamma$ may be related to the estimated standard deviation $\sigma$.

Assuming the target data samples $X_1, ..., X_n$ are i.i.d. with unknown mean $\mu$ and variance $\sigma^2$, we have the sample mean as $\bar{X}_n = \frac{1}{n} \sum_{i=1}^{n} X_i$. By the Central Limit Theorem (CLT), for large enough $n$, we have $\bar{X}_n \sim \mathcal{N}(\mu, \frac{\sigma^2}{n})$ and the standard error of the mean as $\text{SE}(\bar{X}_n) = \frac{\sigma}{\sqrt{n}}$.

To ensure the $(1 - \alpha)$ confidence interval of $\mu$ has width no more than $\varepsilon$:

$$2z_{\alpha/2} \cdot \frac{\sigma}{\sqrt{n}} \leq \varepsilon, \tag{15}$$

where we require:

$$n \geq \left(\frac{2z_{\alpha/2}\sigma}{\varepsilon}\right)^2. \tag{16}$$

In contrast, estimating the variance $\sigma^2$ (as a proxy for $\gamma^2$ in LayerNorm) converges more slowly. Under the assumption that the data follows a Gaussian distribution, the sample variance $\hat{\sigma}^2$ satisfies:

$$\frac{(n-1)\hat{\sigma}^2}{\sigma^2} \sim \chi^2_{n-1}. \tag{17}$$

Hence, the $(1 - \alpha)$ confidence interval for $\sigma^2$ is:

$$\left[\frac{(n-1)\hat{\sigma}^2}{\chi^2_{1-\alpha/2}}, \frac{(n-1)\hat{\sigma}^2}{\chi^2_{\alpha/2}}\right]. \tag{18}$$

To ensure the confidence interval width is within $2\varepsilon$, we solve:

$$\frac{(n-1)\hat{\sigma}^2}{\chi^2_{\alpha/2}} - \frac{(n-1)\hat{\sigma}^2}{\chi^2_{1-\alpha/2}} \le 2\varepsilon. \tag{19}$$

This yields:

$$n \ge 1 + \frac{\sigma^2}{\varepsilon} \cdot \frac{\Delta_\chi}{2}, \quad \text{where } \Delta_\chi = \frac{1}{\chi^2_{1-\alpha/2}} - \frac{1}{\chi^2_{\alpha/2}}. \tag{20}$$

It is worth noting that if the underlying data distribution deviates from Gaussian, a substantially larger sample size may be required to reliably estimate the variance—and hence stabilize $\gamma$.

**Proof details of Proposition 3.1.** We want to highlight the credit of (Kumar et al., 2022) as most of the proof follows it, but several details are different.

Since $Z^T X^\top X Z^{T^\top}$ is invertible for most cases, there is a unique global minimum over a predictor's parameters $v$ to the loss optimized by linear-probing:

$$\arg\min_v \left\| XZ^{T^\top} v - XZ^{S^\top} v^S \right\|_2^2 = \left( Z^T X^\top X Z^{T^\top} \right)^{-1} Z^T X^\top X Z^{S^\top} v^S. \tag{21}$$

Assuming $\mathcal{Z}$ is a Reproducing Kernel Hilbert Space (RKHS), the loss function thus is strongly convex in $v$ since the Hessian $Z^T X^\top X Z^{T^\top}$ is invertible. Then, the minima are unique for the gradient flow convergence in the RKHS for the parameters $v_{\text{lp}}$ of $\mathcal{C}^{T'}$, where:

$$v_{\text{lp}}^\infty = \left( Z^T X^\top X Z^{T^\top} \right)^{-1} Z^T X^\top X Z^{S^\top} v^S, \tag{22}$$

where the $v^S$ is the parameters of the unavailable $\mathcal{C}^S$.

We thus have the following definition:

$$\begin{aligned}
\sqrt{L\left(v_{\text{lp}^{T'}}, Z^T\right)} &= \left\| Z^{S^\top} v^S - B^{T^\top} v_{\text{lp}}^\infty \right\|_2 \\
&\le \left\| \left( Z^{S^\top} v^S - Z^{T^\top} v^S \right) + \left( Z^{T^\top} v^S - Z^{T^\top} v_{\text{lp}}^\infty \right) \right\|_2 \\
&\le \underbrace{\left\| Z^{S^\top} v^S - Z^{T^\top} v^S \right\|_2}_{(1)} + \underbrace{\left\| Z^{T^\top} v^S - Z^{T^\top} v_{\text{lp}}^\infty \right)\right\|_2}_{(2)}.
\end{aligned} \tag{23}$$

For term (1):

$$\left\| B_S^\top v^S - B^{T^\top} v^S \right\|_2 \le \sigma_{\max}\left(B_S - B^T\right) \left\|v^S\right\|_2 \le \epsilon \left\|v^S\right\|_2 = \epsilon \left\|w_\star\right\|_2. \tag{24}$$

Where we note that $\left\|v^S\right\|_2 = \left\|w_\star\right\|_2$ and $w_\star = B_S^\top v^S$ where the rows of $B_S$ (columns of $B_S^\top$) are orthonormal.

Let $\Sigma = X^\top X$. For term (2), we first substitute $v_{\text{lp}}^\infty$ and do some algebra (again noting that $\left\|v^S\right\|_2 = \left\|w_\star\right\|_2$) to get:

$$\begin{aligned}
\left\| B^{T^\top} v^S - B^{T^\top} v_{\text{lp}}^\infty \right\|_2 &= \left\| B^{T^\top} \left( B^T \Sigma B^{T^\top} \right)^{-1} B^T \Sigma \left( B^T - B_S \right)^\top v^S \right\|_2 \\
&\le \sigma_{\max}\left( B^{T^\top} \left( B^T \Sigma B^{T^\top} \right)^{-1} B^T \Sigma \right) \sigma_{\max}\left( B^T - B_S \right) \left\|w_\star\right\|_2 \\
&\le \frac{\sigma_{\max}\left(B^T\right)^2 \sigma_{\max}(X)^2}{\sigma_{\min}\left(XB^{T^\top}\right)^2} \sigma_{\max}\left(B^T - B_S\right) \left\|w_\star\right\|_2 \\
&\le \frac{\sigma_{\max}\left(B^T\right)^2 \sigma_{\max}(Z)^2}{\sigma_{\min}(Z)^2 \left(\cos\theta_{\max}(R,S)\right)^2} \sigma_{\max}\left(B^T - B_S\right) \left\|w_\star\right\|_2.
\end{aligned} \tag{25}$$

Since $B^T$ has orthonormal rows, $\sigma_{\max}\left(B^T\right) = 1.\frac{\sigma_{\max}(Z)^2}{\sigma_{\min}(Z)^2} >= 1$. So it suffices to bound the quantities in the RKHS.

### A.5 TOY EXAMPLE DETAILS

**Toy example training details.** To maintain simplicity, we employ the model with two MLPs and one LayerNorm layer in between. Following the standard fine-tuning process, our toy experiments also conduct training on the source data, fine-tuning on the target training set, and then testing on the rest of the target data. Specifically, the pretriaming stage trains all layers of the model; the fine-tuning stage only tunes the LayerNorm layer. After finetuning, we conduct the $\gamma$ rescaling on the tuned model. All examples are conducted for classification, and accuracy (acc) is used as the evaluation metric. Note that the reported accuracy results are in the range of $[0, 1]$. After fine-tuning, each model's LayerNorm $\gamma$ values are rescaled by $\lambda \in [0, 2]$, and the optimal value of $\lambda$ that leads to the best performance is chosen. All toy examples run on the CPU. The default training seed is set to $42$.

**Toy data generation** To simulate the source domain $X^S$, target training set $X^T$, and testing target sets $X^{T*}$ with class labels, we synthetically generate datasets based on Gaussian distributions with controlled variations in means and variances across multiple classes. Specifically, the source domain data are sampled from class-specific Gaussian distributions centered at fixed means with unit variance. Each distribution contributes an equal number of samples (i.e., 100 samples per class), with labels assigned according to the corresponding Gaussian component. To construct the target domain, we introduce domain shift by perturbing both the means and variances of the source distributions. Mean shifts are applied using predefined displacement vectors scaled by a *mean shift scale* parameter, while variances are modified by scaling the original values using class-specific variance multipliers and a *variance shift scale* factor. As a result, the target distributions exhibit both translational and scale shifts relative to their source counterparts, enabling controlled evaluation of model robustness under varying degrees of domain shift. Similar to the source domain, 100 labeled samples per class are generated for the target domain. A portion of these labeled target samples is used for training, while the remainder is reserved for testing.

We consider three settings for the number of classes: 2, 4, and 8. In each setting, source samples are drawn from fixed Gaussian distributions, while target samples are drawn from the corresponding shifted distributions. The mean shift scale and variance shift scale parameters are varied independently within the range $[0, 2]$ in 11 uniformly spaced steps to simulate increasing degrees of domain shift. For each configuration of the class count, shift parameters, and training conditions, the proportion of labeled target training data is varied among $[0.01, 0.05, 0.1, 0.3, 0.5]$, corresponding to $1, 5, 10, 30,$ and $50$ labeled samples per class to ensure that we cover one-shot, few-shot and normal fine-tuning scenarios. For each model, the shifts of $\gamma$ of LayerNorm are multiplied with the $\lambda$ that varies from $[0, 2]$ with 21 uniformly spaced steps.

**More results.** The results of toy examples of different numbers of classes mentioned in the main paper are exhibited in Fig. 2. Moreover, as shown in Fig. 6, the Fine-tuning Shift Ratio (FSR) decreases as the fraction of target samples used for training increases. In parallel, Fig. 7 illustrates that the optimal value of $\lambda$ increases with larger training fractions. Notably, when the available target data is extremely limited (e.g., a fraction of 0.01, corresponding to one sample per class), the learned LayerNorm parameters $\gamma$ and $\beta$ become heavily biased by individual samples, causing the relationship between $\lambda$ and $FSR$ to become less reliable. Detailed visualization can be seen in Fig. 8.

### A.6 MORE DETAILS ABOUT THE PROPOSED METHOD

#### A.6.1 OUR METHOD: MORE DETAILS AND TRICKS

To further enhance LayerNorm fine-tuning, we introduce two additional techniques: (i) increasing the feature dimension before prediction and (ii) applying a lightweight feature augmentation prior to attention pooling. Expanding the feature dimension leads to a latent space where features become more separable, facilitating a more stable optimization process. Additionally, we introduce a learnable scaling factor for the features before attention pooling, providing mild feature augmentation to promote better generalization.

#### A.6.2 MORE RESULTS

**Ablation study of proposed tricks.** The results in Fig. 9 illustrate the effects of each proposed trick. Here, C1 denotes the baseline without feature dimension expansion, and C2 denotes doubling

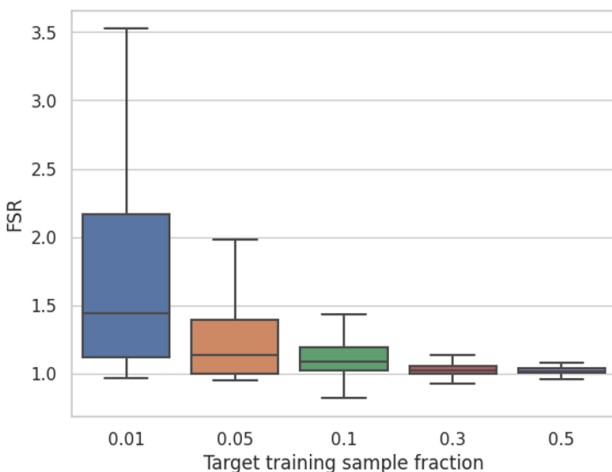

Figure 6: Toy example results: Fine-tuning Shift Ratio ($FSR$) against fractions of target samples used for training.

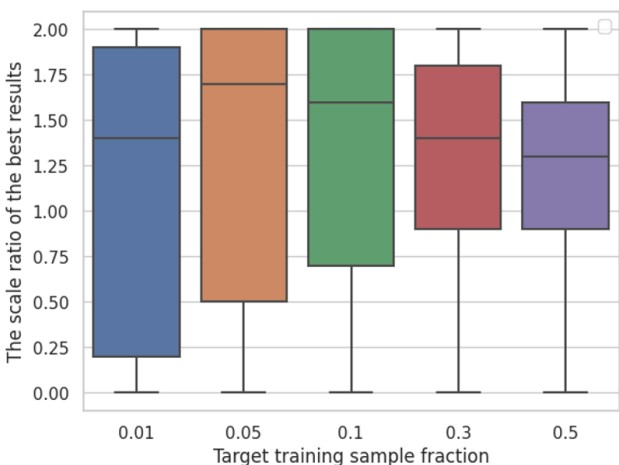

Figure 7: Toy example results: The scaling factor $\lambda$ against fractions of target samples used for training.

the feature dimension. As the results indicate, our training pipeline already significantly boosts performance, and these two additional tricks provide further slight improvements.

**More comparisons of different LayerNorm fine-tuning approaches.** We offer more comparisons of different LayerNorm fine-tuning approaches in Fig. 10. It can be seen that our method yields the best or second-best results under most scenarios, bringing on average the best results among all competitors.

### A.6.3 EXPERIMENTAL DETAILS AND MORE RESULTS FOR NATURAL IMAGE VITFs

**Experimental settings.** For MAE, we pretrain the model by ourselves and then fine-tune it. For the IP setting, the target domain is identical to the source domain, and fine-tuning is performed using labeled data from the same domain. In contrast, for the OOP setting, the target domains consist of the remaining domains not used during pretraining. To systematically analyze performance under varying data regimes, we fine-tune using randomly sampled $1\%$, $5\%$, $10\%$, and $50\%$ fractions of the labeled data from the target domain, while reserving the remaining data for evaluation.

For large-scale pretrained CLIP and DINOv2, as it is difficult to determine whether it is IP or OOP settings, we tune them by using the subsets of overall domains, but test them across all domains. Here,

Table 9: Experimental details for the SUN dataset: Our method for CLIP

| Training fraction | Learning rate | Epochs | Switch epochs | Weight decay |
|---|---|---|---|---|
| All | $5e-4$ | 10 | 5 | $1e-3$ |
| 10% | $5e-4$ | 10 | 5 | $1e-5$ |
| 5% | $1e-3$ | 10 | 2 | $1e-3$ |
| 2% | $3e-3$ | 10 | 2 | $5e-3$ |

Table 10: Experimental details for the SUN dataset: Our method for DINOv2

| Training fraction | Learning rate | Epochs | Switch epochs | Weight decay |
|---|---|---|---|---|
| All | $1e-4$ | 20 | 5 | $1e-4$ |
| 10% | $2e-4$ | 20 | 5 | $1e-4$ |
| 5% | $2e-4$ | 20 | 5 | $1e-4$ |
| 2% | $2e-3$ | 10 | 2 | $5e-3$ |

we further test the large-scale pertaining ViTFs, OpenCLIP (Radford et al., 2021) and DINOv2 (Oquab et al., 2023), with LayerNorm fine-tuning on the domains of the DomainNet dataset used by MAE, SUN (Xiao et al., 2010), DTD (Cimpoi et al., 2014), and Bach (Aresta et al., 2019). Like DomainNet for MAE, we also use randomly sampled 2%, 5%, 10%, and 100% fractions of the labeled data from the target domain for both datasets and the official testing set for evaluation. Specifically, we use the official testing set for the performance evaluation. $\lambda$ is searched in the range of $[0, 2]$ and the best $\lambda$ is chosen by the evaluation on the testing set. All experiments use one A100 GPU with 80GB of memory. The default training seed is set to 42.

### A.6.4 Experimental details for DomainNet dataset

**Pretraining MAE.** Following (He et al., 2022), we adopt the self-supervised ViT-B/16 model as the backbone network. The pretriaining stage uses the learning rate of $1 \times 10^{-4}$ for 500 epochs. The mask ratio is set as 0.80. We adopted the vit_base_patch16_224_in21k as the initialization of the MAE.

**Fine-tuning MAE.** For fine-tuning, all approaches use the learning rate fixed at $5 \times 10^{-4}$, and all models are trained for a total of 100 epochs. Our method performs each tuning round 20 epochs and then switches to another round. The default batch size is 128, but for extremely low data fractions (e.g., 0.01), it is reduced to 12 to account for the scarcity of training samples.

**Fine-tuning CLIP and DINOv2.** For all experiments, the learning rate is fixed at $1 \times 10^{-3}$, and training is conducted for a total of 20 epochs. The batch sizes are kept the same as those used in MAE fine-tuning. For our proposed method, the switch epoch is set to 2.

### A.6.5 Experimental details for the SUN dataset

**Fine-tuning CLIP and DINOv2.** For all experiments, the learning rate is fixed at $5 \times 10^{-4}$ and weight decay is $1 \times 10^{-4}$, and training is conducted for a total of 20 epochs. The batch sizes are 64 for all experiments. Especially for our method on CLIP and DINOv2 fine-tuning, please refer to Tables 9 and 10.

### A.6.6 Experimental details for the DTD dataset

**Fine-tuning CLIP and DINOv2.** For all experiments, the learning rate is fixed at $5 \times 10^{-4}$ and weight decay is $1 \times 10^{-4}$, and training is conducted for a total of 20 epochs. The batch sizes are 64 for all experiments. Especially for our method on CLIP and DINOv2 fine-tuning, please refer to Tables 11 and 12.

### A.6.7 Experimental details for the Bach dataset

**Fine-tuning CLIP and DINOv2.** The experimental details are the same as the pathological ViTFs.

Table 11: Experimental details for the DTD dataset: Our method for CLIP

| Training fraction | Learning rate | Epochs | Switch epochs | Weight decay |
|---|---|---|---|---|
| All | $5e-4$ | 10 | 5 | $1e-3$ |
| 10% | $5e-4$ | 10 | 5 | $1e-5$ |
| 5% | $1e-3$ | 10 | 2 | $1e-3$ |
| 2% | $3e-3$ | 10 | 2 | $5e-3$ |

Table 12: Experimental details for the SUN dataset: Our method for DINOv2

| Training fraction | Learning rate | Epochs | Switch epochs | Weight decay |
|---|---|---|---|---|
| All | $1e-4$ | 20 | 5 | $1e-6$ |
| 10% | $1e-3$ | 20 | 5 | $1e-4$ |
| 5% | $1e-3$ | 20 | 5 | $1e-4$ |
| 2% | $1e-3$ | 10 | 2 | $5e-3$ |

### A.6.8 MORE RESULTS

**Applying $\lambda$ to $\gamma$ is more stable than $\beta$.** Please refer to the Fig. 11.

## A.7 EXPERIMENTAL DETAILS AND MORE RESULTS FOR PATHOLOGICAL VITFS

### A.7.1 EXPERIMENTAL DETAILS

We evaluate the fine-tuning performance of pathological ViTFs (CONCH (Lu et al., 2024), CHEIF (Wang et al., 2024), and TITAN (Ding et al., 2024)) using five benchmark datasets: Bach (Aresta et al., 2019), Breakhis (Spanhol et al., 2015), CCRCC (Brummer et al., 2023) (including both binary and tissue subsets), and Glas (Sirinukunwattana et al., 2017). For each dataset, we randomly split the labeled data in half, using one half exclusively for testing. The target training samples are randomly drawn from the remaining half. Specifically, we experimented with 1, 5, 10, and 20 samples per class, and we additionally included experiments using all available target training samples for Bach and Glas. A fixed random seed 42 is used for all data splits. The experimental settings and baseline methods remain consistent with those used for natural image datasets, except for the omission of LP+WM tuning due to its consistently poor performance, particularly in limited target training samples scenarios, as demonstrated in Tables 2 and 7. Similar to natural image ViTFs, $\lambda$ is further chosen based on test set performance as the indicator of $FSR$.

For a fair comparison, we train all ViTFs using the seed 1 as the default. All experiments across all datasets and ViTFs are using 0.001 as the learning rate with 100 epochs. Especially for our method, we set the epoch of each round as 20, i.e., switching the training stage at every 20 rounds. All experiments use one A100 GPU with 80GB of memory. The default training seed is set to 42.

### A.7.2 MORE RESULTS

**Detailed results of segmentation.** Detailed results of Table 4 segmentation of each class can be seen in Table 13.

**More results of pathology ViTFs** can be found in Figs. 13 and 14. More visualizations of the LayerNorm shifts can be seen in Fig. 12.

**Stability of our proposed method.** We provide boxplots of our proposed method across various seeds in Fig. 15. Moreover, we also test different fine-tuning methods across seeds $[0, 1, 2]$ Table 14. It can be seen that our method yields stable improvements with relatively smaller variances across all settings. This indicates the robustness of our proposed method.

Table 13: IP and OOP: Results for segmentation adaptation with various methods (using officially released ckpt). Ro: road; SW: sidewalk; B: building; W: wall; F: fence; P: pole; TL: traffic light; TS: trafficsign; V: vegetation; Te: terrain; Sky: sky; P: person; R: rider; C: car; T: truck; Bus: bus; Tra: train; Mo: motorcycle; Bi: bicycle.

| Class | mAcc | Ro | SW | B | W | F | P | TL | TS | V | Te | Sky | P | R | C | T | Bus | Tra | Mo | Bi |
|---|---|---|---|---|---|---|---|---|---|---|---|---|---|---|---|---|---|---|---|---|
| ID: Segformer (CS->CS) | | | | | | | | | | | | | | | | | | | | |
| Acc | 96.73 | 99.16 | 93.31 | 97.0 | 76.94 | 74.82 | 81.24 | 87.0 | 89.37 | 96.89 | 76.34 | 98.15 | 92.84 | 84.57 | 98.19 | 89.3 | 96.14 | 90.69 | 83.69 | 89.89 |
| +$\lambda$=0.95 | 96.75 | 99.18 | 93.24 | 97.01 | 78.34 | 74.75 | 81.34 | 85.74 | 88.61 | 96.97 | 77.02 | 98.18 | 92.51 | 84.32 | 98.11 | 89.13 | 95.84 | 90.69 | 82.35 | 89.9 |
| OOP: Segformer+DAformer (Synthia->CS) | | | | | | | | | | | | | | | | | | | | |
| Acc | 61.54 | 86.44 | 70.92 | 94.81 | 55.59 | 9.59 | 59.69 | 66.16 | 63.05 | 95.31 | - | 98.86 | 87.88 | 67.32 | 95.64 | - | 85.92 | - | 60.38 | 71.68 |
| +$\lambda$=1.05 | 61.64 | 83.27 | 71.4 | 94.79 | 54.92 | 10.01 | 60.13 | 67.51 | 64.51 | 94.95 | - | 98.85 | 88.08 | 66.53 | 95.74 | - | 86.45 | - | 63.15 | 70.84 |

Table 14: Averaged results with standard deviations of different fine-tuning methods across seeds $[0, 1, 2]$ of MAE on DomainNet domain splits.

| Finetune to A | test on A | | | test on B | | | Finetune to B | test on B | | | test on A | | |
|---|---|---|---|---|---|---|---|---|---|---|---|---|---|
| | Lora | LP-LN | Ours | Lora | LP-LN | Ours | | Lora | LP-LN | Ours | Lora | LP-LN | Ours |
| 1% | 73.79 | 82.93 | 86.87 | 43.55 | 55.03 | 60.11 | 1% | 45.00 | 60.30 | 69.62 | 61.41 | 66.35 | 77.41 |
| *Std.* | *3.04* | *3.59* | *1.03* | *0.39* | *6.65* | *1.20* | *Std.* | *1.30* | *10.82* | *1.36* | *10.98* | *10.02* | *1.75* |
| 10% | 92.38 | 91.91 | 93.23 | 65.53 | 64.25 | 66.23 | 10% | 80.13 | 81.01 | 81.94 | 87.87 | 83.51 | 89.12 |
| *Std.* | *0.18* | *0.30* | *0.21* | *0.87* | *0.49* | *0.75* | *Std.* | *0.28* | *0.34* | *0.65* | *0.26* | *0.77* | *0.53* |

### A.7.3 ATTEMPTS TO SPARSIFY THE LAYERNORM SHIFTS UNDER LIMITED TARGET TRAINING SAMPLES

As observed in Section 4.4, increasing the number of tuning samples generally results in sparser LayerNorm updates and improved performance. We have attempted to achieve sparsity under limited target training samples to enhance performance further. However, due to the randomness in the training process and the selection of target samples, achieving an appropriate level of sparsity seems to be non-trivial. We present our attempts here, hoping that these results may benefit those interested in further exploring this direction.

**Baseline.** We use the LP results as the baseline.

**SVD Keep first for gamma only.** All shifts in $\gamma$ parameters are concatenated into a single matrix, with columns corresponding to feature dimensions and rows corresponding to different layers. Singular Value Decomposition (SVD) is then applied to this matrix, and the top-$k$ singular components are retained. The reported results correspond to the optimal choice of $k$ selected based on validation performance.

**SVD Keep last for both gamma and beta.** All shifts in $\gamma$ and $\beta$ parameters are concatenated into a single matrix correspondingly, with columns corresponding to feature dimensions and rows corresponding to different layers. Singular Value Decomposition (SVD) is then applied to both matrices, and the last-$k$ singular components are retained. The reported results correspond to the optimal choice of $k$ selected based on validation performance.

**SVD Keep middle for both gamma and beta.** All shifts in $\gamma$ and $\beta$ parameters are concatenated into a single matrix correspondingly, with columns corresponding to feature dimensions and rows corresponding to different layers. Singular Value Decomposition (SVD) is then applied to both matrices, and the middle-$k$ singular components are retained. The reported results correspond to the optimal choice of $k$ selected based on validation performance.

**SVD Keep first for beta only.** All shifts in $\beta$ parameters are concatenated into a single matrix, with columns corresponding to feature dimensions and rows corresponding to different layers. Singular Value Decomposition (SVD) is then applied to this matrix, and the top-$k$ singular components are retained. The reported results correspond to the optimal choice of $k$ selected based on validation performance.

**SVD Keep first for both gamma and beta.** All shifts in $\gamma$ and $\beta$ parameters are concatenated into a single matrix correspondingly, with columns corresponding to feature dimensions and rows corresponding to different layers. Singular Value Decomposition (SVD) is then applied to both matrices, and the first-$k$ singular components are retained. The reported results correspond to the optimal choice of $k$ selected based on validation performance.

**Random drop for gamma only.** Randomly drop shifts in $\gamma$ by ratios. The reported results correspond to the optimal choice of the drop ratios selected based on validation performance.

**Random drop for beta only.** Randomly drop shifts in $\beta$ by ratios. The reported results correspond to the optimal choice of the drop ratios selected based on validation performance.

**Only scale beta.** Rescaling shifts in $\beta$. The reported results correspond to the optimal choice of the rescale ratios selected based on validation performance.

**Only scale gamma (Ours).** Rescaling shifts in $\gamma$. The reported results correspond to the optimal choice of the rescale ratios selected based on validation performance.

**Result.** As shown in Fig. 16, Only scale gamma (Ours) achieves the best results. Moreover, we find that the SVD-based approaches often fail to bring improvements when there are more training samples (e.g., 20 per class) and may not be able to be effective for all modes (e.g., CHEIF). Thus, while sparsifying LayerNorm updates in low-data regimes may offer potential benefits, it remains a non-trivial challenge due to the complex, model-specific nature of these adaptations and the inherent uncertainty associated with the selected target training samples.

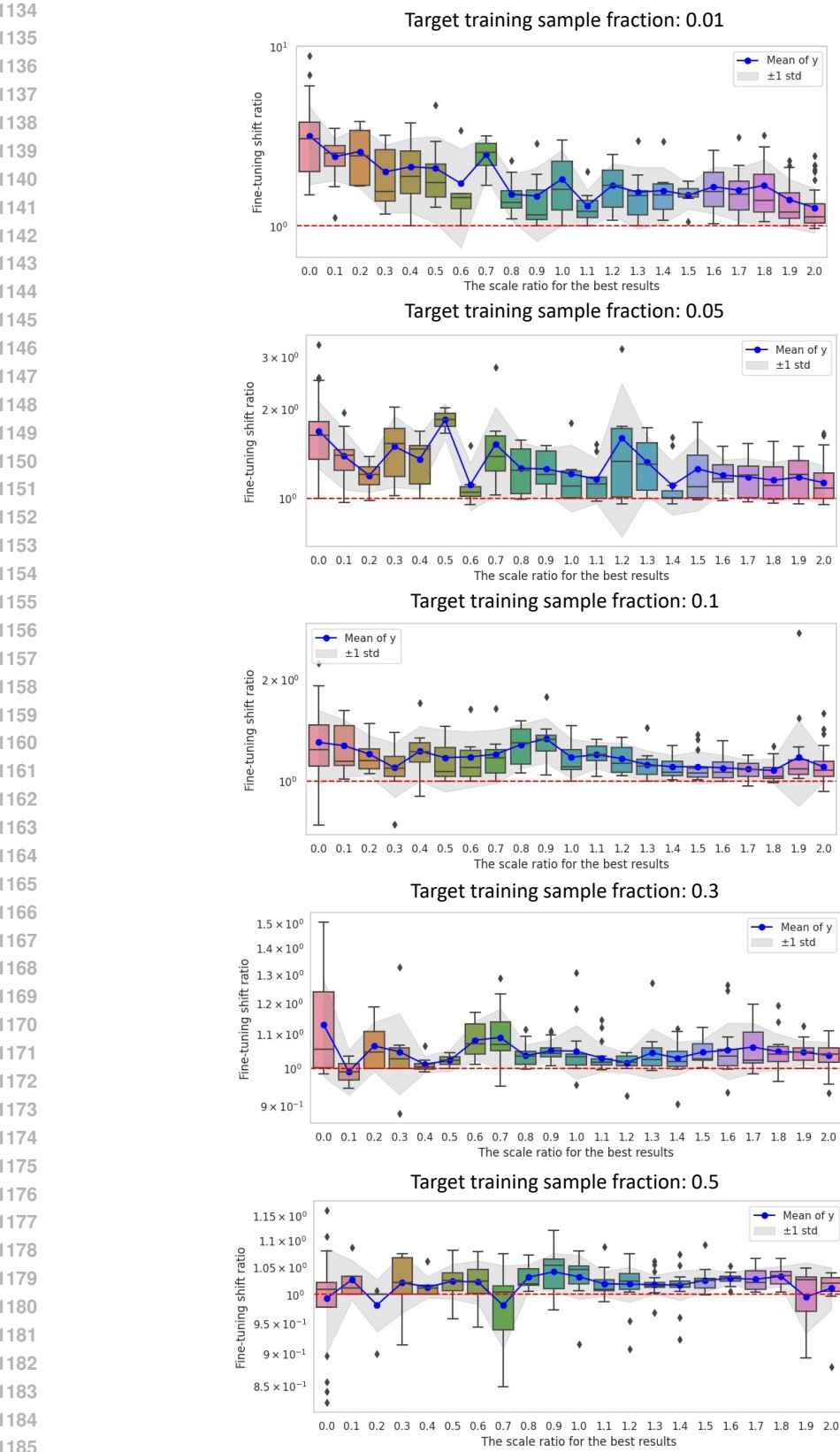

Figure 8: Toy example results: Statistical visualization of the relationship between the Fine-tuning Shift Ratio ($FSR$) and the scaling factor $\lambda$ that achieves the best performance, evaluated across varying fractions of target samples used for training.

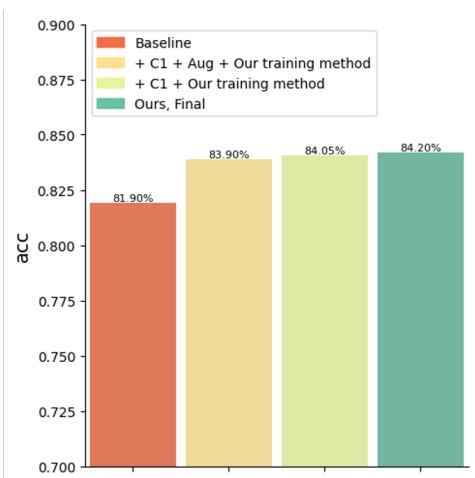

Figure 9: Ablation of proposed tricks of CONCH tuned on Bach. The results are averaged across various target training sample fractions.

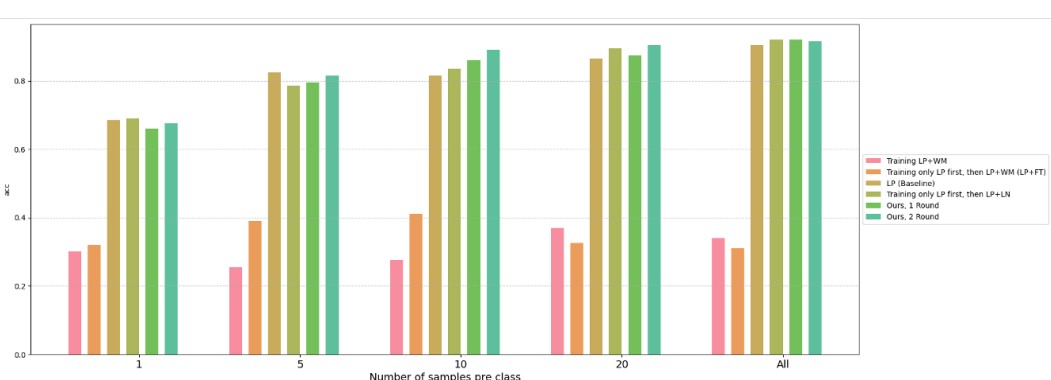

Figure 10: Our proposed method compared with various LayerNorm fine-tuning approaches. The used ViTF is TITAN, and the used dataset is Bach.

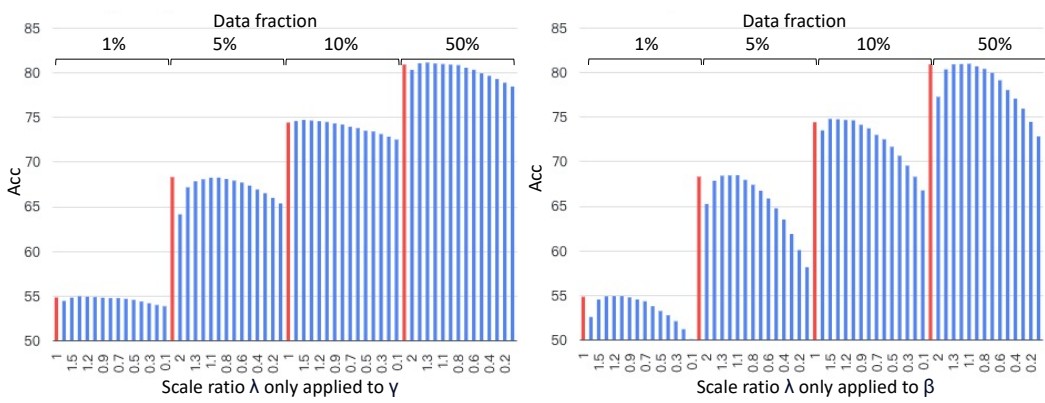

Figure 11: Comparison of applying $\lambda$ to $\gamma$ or $\beta$. The model adopted here corresponds to the IP:A to A experiments from Table 7. Though the tendencies are similar, it can be seen that the $\beta$ is way more sensitive to the value of $\lambda$ than $\gamma$. This reflects the analysis in Section 2.1. Therefore, we only apply the $\lambda$ to $\gamma$.

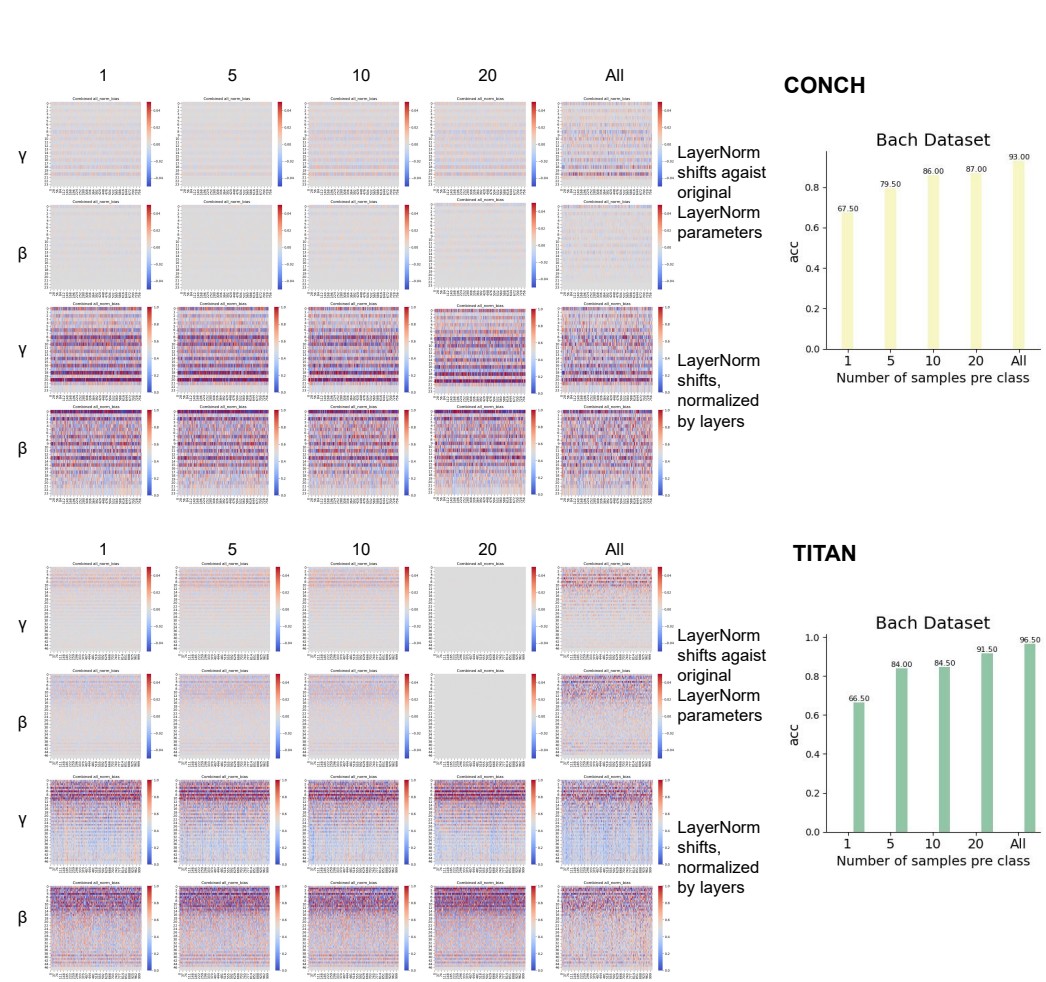

Figure 12: Visualizations of LayerNorm shifts on the Bach dataset for each setting.

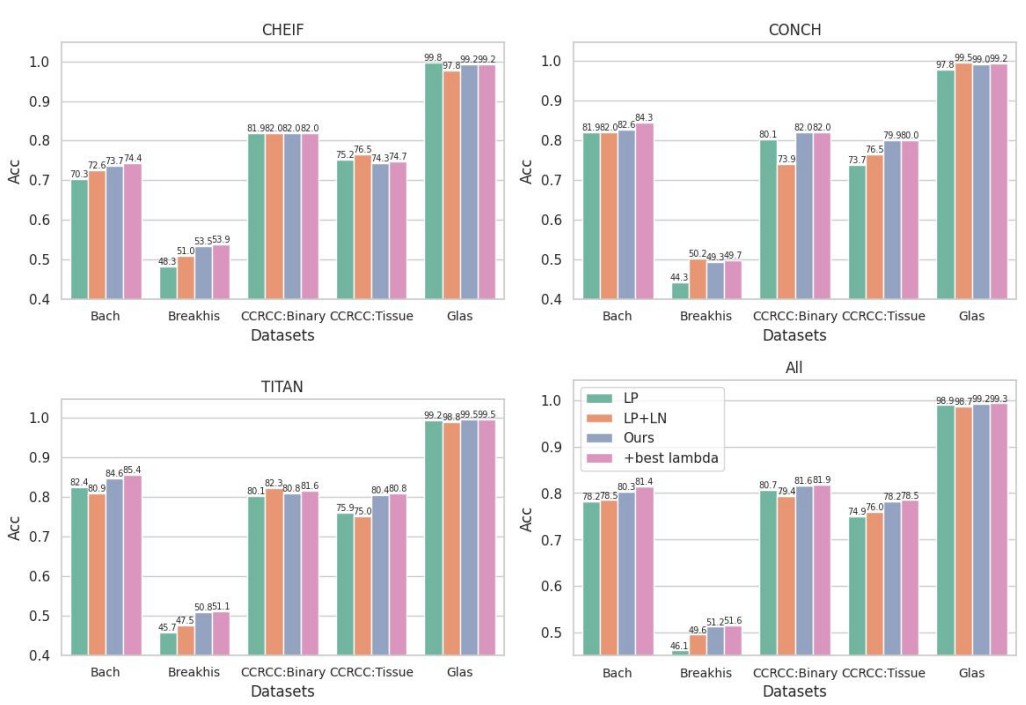

Figure 13: Results of fine-tuning pathological foundation models across datasets. Note here that all results are averaged across different numbers of target training samples per class.

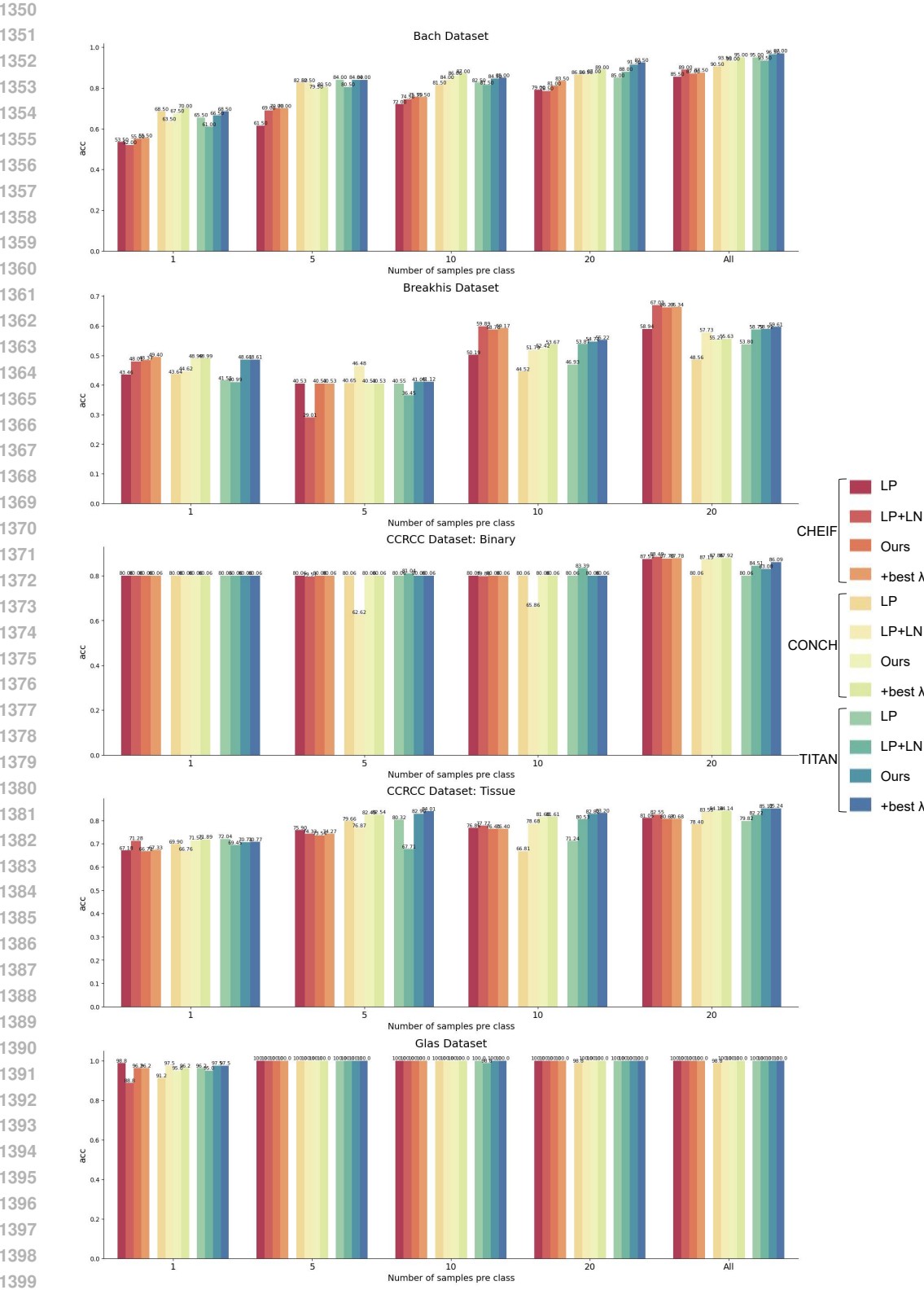

Figure 14: Results of each pathological dataset against different numbers of target training data for each class.

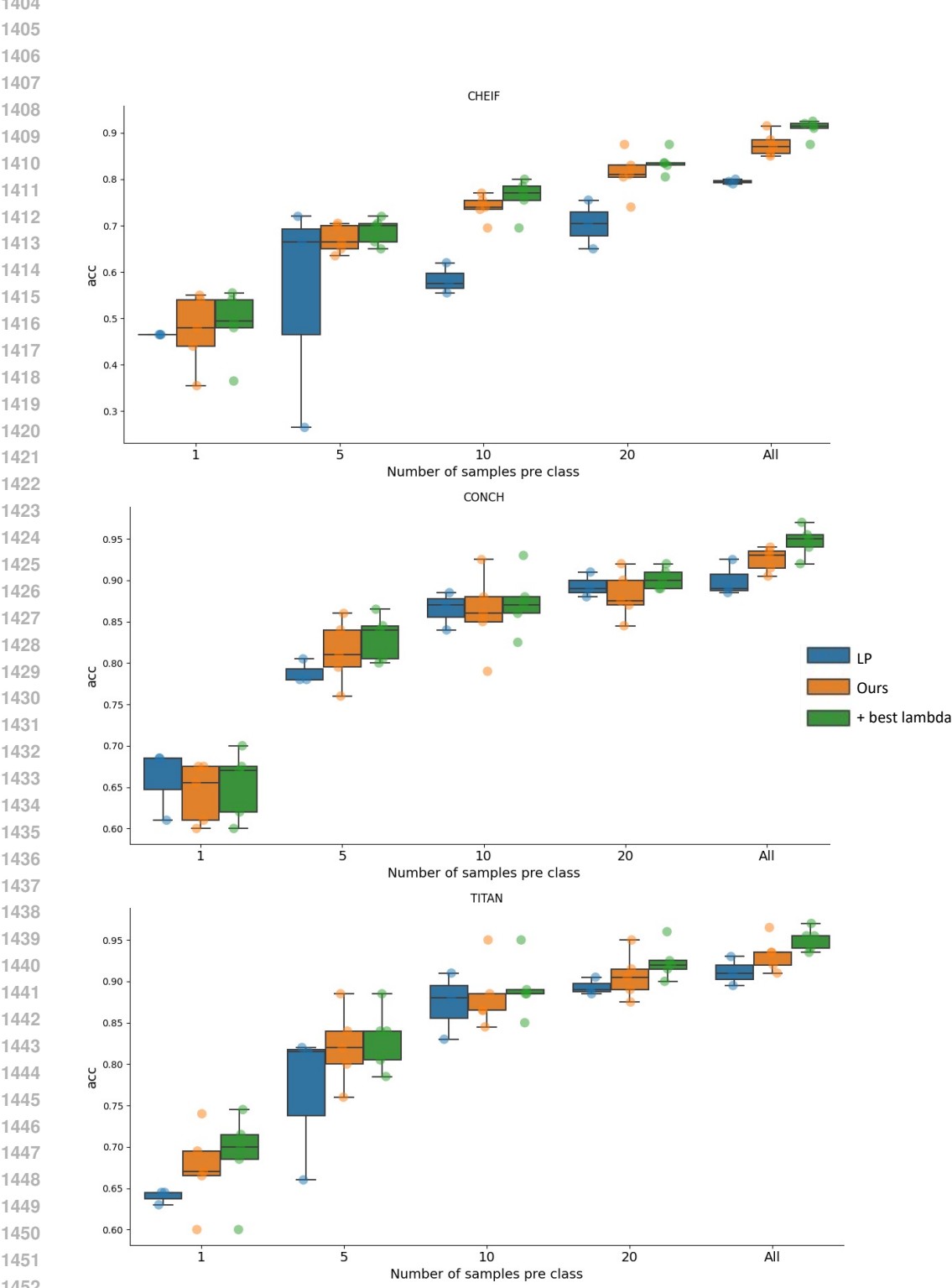

Figure 15: Our proposed method across various seeds for data splits in comparison to LP. The seeds used here are $[1, 2, 3, 4, 42]$.

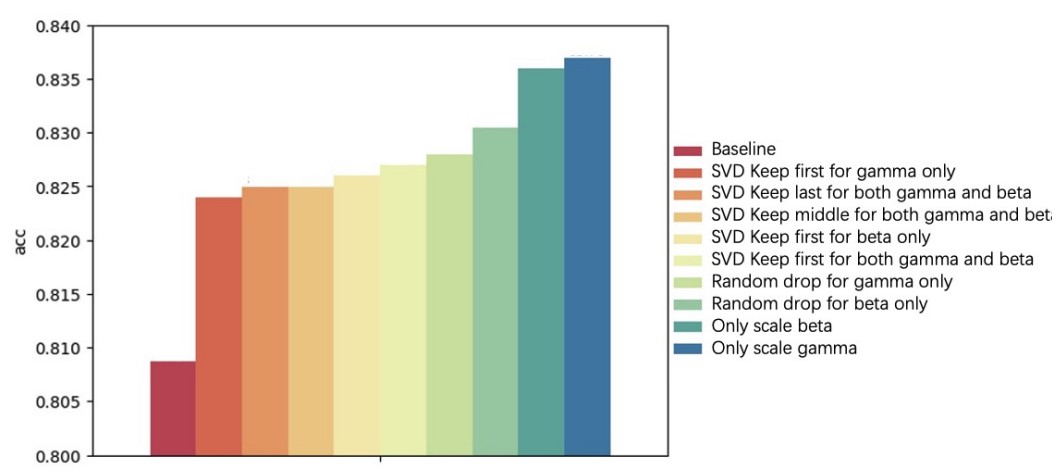

Figure 16: Different LayerNorm sparsity attempts. The used dataset is Bach. The results are averaged across all factions and ViTFs.

