# OpenReview forum: "Is Layer Normalization Fine-tuning Sufficient for Visual Distribution Shifts?"
_ICLR.cc/2026/Conference — Submitted to ICLR 2026_

### Official Review · Reviewer_8NnW · 2025-10-18

**Soundness:** 1
**Presentation:** 1
**Contribution:** 1
**Rating:** 0
**Confidence:** 5

**Summary:**

The authors claim that data distribution shifts between the pre-training (source) and fine-tuning (target) datasets can be captured by the parameter space l2-norm shift in the layer-normalization layer. Then, they propose a fine-tuning shift ratio (FSR), which is constructed by source, target, and optimal target data distribution, to derive a formulation to rescale the LayerNorm mean parameter. They validate their method on a Gaussian synthetic dataset with MLP, and some real image datasets with OpenCLIP and DINOv2.

**Strengths:**

There are no substantive strengths in terms of originality, quality, clarity, and significance. Please see the weaknesses section.

**Weaknesses:**

* **Significant issue with the writing quality**
  * Inconsistent terminology usage
    * line 012: Vision Transformer Foundation models (ViTFs) v.s. line 030: visual foundation models (ViTFs)
    * the $n$ is denoted as a number of layers in line 079 and provable Eq (3) as well, but used as a number of samples in line 170.
  * Tons of typos and wrong statements (even in the most important formulation)
    * line 050: ViF's fine-tuning -- must be ViT's fine-tuning
    * line 159: they assume the scenario $FSR < 1$ or $FSR > 1$, but they derive the Eq (6) with $FSR=1$
    * Eq (7): the denominator and numerator are the same. This will be equal to 1. Maybe the $T$ in the denominator should be $T^{*}$
    * line 478: ID/OOD -- must be IP/OOP
    * line 701: the sentence "where $\mathcal{M}_{\mathcal{LN}}$ is the $\mathcal{LN}$ in the model without and ..." is incorrect. Without what?
  * Inprofessional notation usage and underspecified sentences (even in the most important formulation)
    * line 095: using $Shift_{ln}$ and $Shift_{data}$ far before defining it explicitly.
    * line 171: sample mean of which quantities? converge to where? Please clarify these two.
    * line 177: where is the definition of $\mu$ and $\sigma$?
    * Eq. (8): The authors defined the $Shift_{ln}()$ function as a summation of l2 distances between learned parameters. Multiplying $\lambda$ by this shift function can not be expressed as Eq. (8), as the l2 norm is defined with a square root, not a linear sum.
  * Others
    * line 086: please separate $\mathcal{M}^{S}$ $\mathcal{M}^{T}$ and $\mathcal{LN}^{S}$ $\mathcal{LN}^{T}$ with comma (,).
    * Proposition 2.1. line 096-098: unnecessarily verbose description in the theorem. It would be great to move these two descriptive
sentences out to proposition 2.1.
    * line 117: "which can obscure or even degrade the ability of LayerNorm ..." -> this sentence is not connected with the previous sentences 115-116, which claim non-LayerNorm layers' entanglement to task-specific factors.
* **Claims without justification/validation/proofs**
  * line 118-119: "_LayerNorm primarily captures distributional variations in a more disentangled manner, focusing on normalization statistics without encoding additional task-specific knowledge._" -- There is no evidence (citation or validation from authors themselves) that supports this claim. Indeed, the relationship between model components and their capability to capture distribution shifts can be more related to the depth of components rather than specific units (Lee et al. 2022).
  * line 179: the authors say "Since $\beta$ hasn't converged, ...", but how do we know whether $\beta$ is converged or not (to where)? Please provide supporting evidence on this design rationale statement.
* **Flaw in Theory (proposition 2.1.), and low overall formulation quality**
  * The proof of Proposition 2.1. is extremely informal, contains incorrect statements, and the conclusion of the proof does not assure the validity of Eq (2).
    * First of all, we can not linearly decompose the loss function as Eq. (12). How did you define the two loss terms (w. and w.o. LayerNorm), respectively? How can we get the loss from the LayerNorm unit only or its negation? Exactly what assumption did you make for this linear loss decomposition to happen?
    * line 712: learning the classifier head (linear prediction) can not affect the data itself, but the authors say that it bridges the gap between $Y^{T}$ and $Y^{S}$. What do they mean by bridge the gap in the label space?
    * line 721-723: This is not true. The left part of Eq. (13) is just about the LayerNorm's loss difference between source and target, which is not equal to the distribution shifts between $X^S$ and $X^T$. Given this, the author's completion of the proof does not assure the validity of the Eq (2) they want to prove.
  * The equality used in Eq. (8) is not actually equality. It should be an approximation or the author's definition. The authors' rescaled version of the LayerNorm shift measure can not be equal to $f(FSR=1)$.
* **Unrealistic distribution shift setup**
  * The authors investigate the setup where we can access the data from the target domain $(X^T, Y^T)$.
  * This is not a typical distribution shift setup (robust fine-tuning; Kumar et al. 2022, Wortsman et al. 2022, or domain adaptation; Farahani et al. 2020), where we can not access the (labeled or even unlabeled) target domain data at all.
* **No significant improvement for the key proposed component and unfair experiment setup**
  * Not only the proposed rescaling LayerNorm, but the authors also use two additional tricks in their fine-tuning framework: (1) increasing feature dimensions before prediction and (2) applying lightweight feature augmentation before attention pooling.
  * However, as we can see from all the results tables, the performance gain from $\lambda$ is very minor (there is even no improvement in some cases -- See Table 2 Second OOP block # of class 20 row), and the major boost seems to be derived from these two additional tricks, which are not the contributions of this work.
* **Missing technical details**:
  * Although the authors mainly discuss the role of LayerNorm and its variant across the whole paper, they use two additional techniques (increasing feature dimension and feature augmentation). However, the authors do not provide any implementation details on these techniques, even in the Appendix A.6.
  * Besides, the authors do not discuss how they decide IP and OOP for OpenCLIP and DINOv2. Why are SUN and DTD recognized as OOP to the OpenCLIP? Based on what criteria?
---

> Reference
- Lee et al. 2022, "SURGICAL FINE-TUNING IMPROVES ADAPTATION TODISTRIBUTION SHIFTS"
- Kumar et al. 2022, "Fine-Tuning can Distort Pretrained Features and Underperform Out-of-Distribution"
- Wortsman et al. 2022, "Robust fine-tuning of zero-shot models"
- Farahani et al. 2020, "A Brief Review of Domain Adaptation"

**Questions:**

See the weakness section.

---

### Official Review · Reviewer_mWra · 2025-10-29

**Soundness:** 2
**Presentation:** 3
**Contribution:** 3
**Rating:** 6
**Confidence:** 4

**Summary:**

This paper investigates whether fine-tuning only LayerNorm (LN) parameters is sufficient for adapting Vision Transformers (ViTs) to distributional shifts. It provides both theoretical and empirical analyses: (1) showing that LN parameter shifts track data distribution shifts; (2) proposing the Fine-tuning Shift Ratio (FSR) as a conceptual measure of data representativeness; and (3) introducing a practical iterative optimization scheme alternating between predictor and LN updates, optionally combined with a γ-scaling factor λ. Extensive experiments across IP (in-pretraining) and OOP (out-of-pretraining) settings, and multiple datasets (natural and pathological), support the findings. Overall, the paper is well-executed, experimentally rich, and offers non-trivial insights into why LayerNorm-only or parameter-efficient fine-tuning works.

**Strengths:**

- **Comprehensive empirical validation**:
The authors evaluate across a broad range of ViT models (OpenCLIP, DINOv2, MAE) and domains (DomainNet, BACH, Synthia → Cityscapes), with consistent ablations. The study systematically bridges theoretical motivation with practical outcomes.

- **Iterative LP–LN training framework**:
The proposed alternating optimization between the predictor and LayerNorm (decoupling label and distributional gradients) clearly improves both stability and generalization. This part provides the most significant and robust performance gain.

- **Conceptual contribution (FSR view)**:
The FSR-based interpretation of data representativeness gives a principled lens to understand when LN fine-tuning succeeds or fails, tying empirical patterns (IP vs OOP) to underlying data properties.

- **Thorough analysis of convergence and variance**:
The distinction between γ and β convergence speeds, and how rescaling affects adaptation stability, adds interpretability beyond raw performance metrics.

**Weaknesses:**

* **Overemphasis on λ-scaling**
  The λ-scaling component is theoretically elegant but contributes marginally (<0.5%) to actual performance gains. The paper’s framing overstates its role relative to the iterative LP–LN training, which is the true source of improvement. Moreover, λ selection is purely empirical—loosely determined by task type (λ<1 for IP, λ>1 for OOP) and highly dataset-dependent.

* **Predictor dependency**
  Although Proposition 3.1 highlights the interference between predictor and LN gradients, the method still requires a learnable predictor for supervision. It remains unclear whether a cosine or prototype classifier (like CLIP) could fully replace the predictor, allowing pure LN tuning. Clarifying whether the “predictor” functions as a *projector* (feature → feature) or a *classifier* (feature → label) would help generalization.

* **FSR observability limitation**
  FSR is an elegant theoretical construct but practically unobservable since (X_{T^*}) is unavailable. It is inferred only via validation performance or LN parameter shift, limiting its predictive value as a measurable indicator.

* **Lack of formal convergence or gradient analysis**
  While the iterative LP–LN optimization performs well empirically, there is no theoretical analysis of its convergence or gradient dynamics (e.g., gradient interference reduction). Also, it would be helpful to explain or analyze how LP updates relate to label information, while LN relates to domain shift from gradient or other perspectives.

**Questions:**

* **Predictor interpretation**
  Could the authors clarify whether their “predictor” serves as a *projector* or true *classifier*? Would a cosine or CLIP-style classifier allow fully predictor-free LN tuning?

* **Adaptive λ-scaling**
  Could λ be replaced with a learned or adaptive mechanism rather than a fixed, empirically chosen scalar?

* **Failure cases**
  Are there clear failure cases (e.g., extreme FSR values or severe domain shifts) where λ-scaling or LN-only tuning collapses?

* **Gradient correlation analysis**
  A quantitative analysis of gradient correlation between LN and predictor updates would strengthen the claim of gradient decoupling.

---

### Official Review · Reviewer_jiiV · 2025-10-30

**Soundness:** 2
**Presentation:** 1
**Contribution:** 2
**Rating:** 4
**Confidence:** 3

**Summary:**

This paper studies LayerNorm fine-tuning under different domain settings and data regimes. Specifically, it introduces the fine-tuning shift ratio (FSR) and investigates how shifts in LayerNorm parameters relate to data shifts in both in-pretraining and out-of-pretraining scenarios. The results suggest that a rescaling factor $\lambda$ is needed to calibrate the variance (i.e., $\gamma$) parameters in LayerNorm. Additionally, the paper proposes an alternating strategy that switches between tuning the classifier and tuning the LayerNorm parameters to further improve fine-tuning performance. Experimental results show that the proposed approach outperforms existing fine-tuning methods that do not introduce extra parameters. Furthermore, incorporating the optimal rescaling factor $\lambda$ also leads to consistent accuracy improvements.

**Strengths:**

- The paper proposes a simple fine-tuning approach that alternates between tuning the classifier and the LayerNorm layers. The simplicity of the method makes it easy to understand and implement.

- The experimental evaluation is thorough, spanning multiple datasets, backbone models, domain settings (in-pretraining and out-of-pretraining), and varying data regimes (2%, 5%, 10%, and 100%).

**Weaknesses:**

- The paper focuses exclusively on LayerNorm-based fine-tuning, which has already been explored extensively in prior work [1]. Furthermore, the experimental comparisons are limited to full fine-tuning, linear probing, and LP-LN, and do not cover the broader family of parameter-efficient fine-tuning approaches. This gap remains even for methods that, like the proposed approach, do not introduce extra trainable parameters [2, 3].

- According to Line 323, $\lambda$ is selected directly based on test performance. However, despite being emphasized and analyzed in the paper, there is no systematic or practical strategy provided for determining $\lambda$ in real-world scenarios. Moreover, the improvement gained from using the optimal $\lambda$ appears to be only marginal compared to the results obtained without it.

- The overall presentation of the experimental results is not very clear. The font sizes in several tables are too small to read, and the tables appear to present all results at once without clear structure or grouping, which makes them difficult to follow. In addition, there are typo errors throughout the paper. For example, in Equation (7), the denominator should be T* instead of T; at Line 180, $\lambda$ should be applied to $\gamma$ rather than $\beta$; and in Figure 5, the left column misspells the word “against.”


[1] 2025 CVPR Lessons and Insights from a Unifying Study of Parameter-Efficient Fine-Tuning (PEFT) in Visual Recognition

[2] 2022 ACL BitFit: Simple Parameter-efficient Fine-tuning for Transformer-based Masked Language-models

[3] 2023 ICCV DiffFit: Unlocking Transferability of Large Diffusion Models via Simple Parameter-efficient Fine-Tuning

**Questions:**

Besides the weakness shown in the above section, please also see the following questions:

Q1: Why does Table 5 indicate that $\beta$ converges faster than $\gamma$? It seems that $\gamma$ is closer to its optimal value than $\beta$ as the number of training samples increases.

Q2: How many iterations are used when alternating between tuning the classifier and tuning the LayerNorm parameters? Additionally, how is this hyperparameter selected?

---

### Official Review · Reviewer_MA6S · 2025-11-01

**Soundness:** 1
**Presentation:** 2
**Contribution:** 2
**Rating:** 2
**Confidence:** 3

**Summary:**

This paper studies how LayerNorm parameters shift during fine-tuning of Vision Transformer foundation models, especially under data scarcity and domain shifts. The authors argue that LayerNorm shifts reflect how well the target domain is represented in fine-tuning data. They introduce the Fine-tuning Shift Ratio (FSR) to measure representation consistency and propose a simple rescaling mechanism (based on FSR) with a cyclic fine-tuning strategy. Experiments on multiple vision benchmarks show correlations between FSR, domain shifts, and fine-tuning success.

**Strengths:**

1. Novel focus on LayerNorm behavior during transfer learning.
2. Proposes a lightweight scaling and cyclic fine-tuning method, validated on various ViT tasks.

**Weaknesses:**

1. Theoretical claims are informal and may be incorrect.
2. Evaluation is narrow: limited to discriminative models and image classification task; no experiments on generative models (e.g., Qwen), nor on other important vision tasks such as object detection, segmentation, or VQA, limiting the claimed generality.

**Questions:**

Please refer to the weaknesses.

---

### Author Response · Authors · 2025-11-25

Dear Reviewers,

Thank you very much for your valuable time and insightful comments on our paper. We have noticed a significant difference in the scores, and we truly appreciate the diverse perspectives, which help us reflect more critically on our work.

To clarify the core contribution of our paper, our main goal is to share an interesting finding regarding the behavior of Layer Normalization (LayerNorm) during fine-tuning in both In-Pretraining (IP) and Out-of-Pretraining (OOP) scenarios. Specifically, **we observed phenomena of overshooting and undershooting in LayerNorm parameter shifts and found that a rescaling mechanism can alleviate these issues to a certain extent**. This discovery, in our view, provides new insights into the dynamics of LayerNorm in transfer learning for visual foundation models.

**We hypothesize that such phenomena may be universally existing across different domains and tasks.** While we have conducted extensive experiments to validate our observations, we acknowledge that these experiments might not cover all possible scenarios. For certain reasons, we may not make further revisions to the current version of the paper. However, we would still be delighted if any reviewer or reader is willing to explore whether similar phenomena exist in their own research domains and engage in further discussions with us. Your verification and feedback would greatly enrich our understanding of this topic.

Once again, we sincerely thank all reviewers for your constructive inputs. We look forward to your continued discussions.

---

### Meta-Review · Area_Chair_5zew · 2026-01-06

**Summary:**

There have been strong issues about the quality of the writing of the paper and there is no real rebuttal about this from the authors. Concerns of other issues are secondary if a paper has poor quality. I am basing this on this factor for my decision

**Reviewer Concerns:**

8NnW and jiiV brought out issues with the paper's presentation with no real rebuttal for it

**Reviewer Scores:**

I dont think any of the scores would have moved given the lack of rebuttal

---

### Decision · Program_Chairs · 2026-01-26

Reject